# Encoding of long-term associations through neural unitization in the human medial temporal lobe

Hernan G. Rey[1], Emanuela De Falco[1], Matias J. Ison [1,2], Antonio Valentin[3,4], Gonzalo Alarcon[3,4,5], Richard Selway[6], Mark P. Richardson[3] & Rodrigo Quian Quiroga[1]

Besides decades of research showing the role of the medial temporal lobe (MTL) in memory and the encoding of associations, the neural substrates underlying these functions remain unknown. We identified single neurons in the human MTL that responded to multiple and, in most cases, associated stimuli. We observed that most of these neurons exhibit no differences in their spike and local field potential (LFP) activity associated with the individual response-eliciting stimuli. In addition, LFP responses in the theta band preceded single neuron responses by ~70 ms, with the single trial phase providing fine tuning of the spike response onset. We postulate that the finding of similar neuronal responses to associated items provides a simple and flexible way of encoding memories in the human MTL, increasing the effective capacity for memory storage and successful retrieval.

[1] Centre for Systems Neuroscience, University of Leicester, Leicester LE1 7RH, UK. [2] School of Psychology, University of Nottingham, Nottingham NG7 2RD, UK. [3] Division of Neuroscience, Institute of Psychiatry Psychology and Neuroscience, King's College London, London SE5 8AF, UK. [4] Department of Clinical Neurophysiology, King's College Hospital NHS Trust, London SE5 9RS, UK. [5] Comprehensive Epilepsy Center, Neuroscience Institute, Academic Health Systems, Hamad Medical Corporation, Doha PO Box 3050, Qatar. [6] Department of Neurosurgery, King's College Hospital NHS Trust, London SE5 9RS, UK. These authors contributed equally: Hernan G. Rey, Emanuela De Falco  Correspondence and requests for materials should be addressed to R.Q.Q. (email: rqqg1@le.ac.uk)

The medial temporal lobe (MTL) has a key role in declarative memory[1–4], which relies on the encoding of associations between items[5–8], as it has been shown with studies in animals[1,9–16] as well as lesion and imaging studies in humans[4,17–20]. The study of such coding is indeed critical for understanding the mechanisms of how memories are stored in the MTL. However, despite decades of research in this area, we still do not know what is the code that underlies the encoding of memories and associations in the MTL.

In humans, a recent study showed that, while subjects learned a pair-association paradigm, neurons that originally responded to a given item expanded their tuning to encode an associated item, but firing with a lower response strength[21]—i.e., the neuron's graded firing was enough to discriminate between the item originally coded by the neuron and the one that was associated with it. Nevertheless, it remains to be determined if such graded responses are maintained once associations are consolidated, or if a different type of code underlies the long-term representation of associations. Long-term associations have been described in the human MTL during passive viewing (i.e., without having subjects performing an associative learning paradigm), with neurons responding preferentially to known and associated stimuli (e.g., two related persons)[22]. However, in that study only 6 presentations per stimulus were used, which in principle does not offer enough statistical power to compare the neural responses between the different items that the neurons fired to.

In this work, we sought to compare the neural responses of well-learned associated stimuli to gain insights on the neural code underlying the long-term representation of associations in the human MTL. For this, we exploited the unique opportunity to record the activity of multiple individual neurons in patients implanted with electrodes in the MTL for clinical reasons[23]. Specifically, we designed an experiment where we first identified stimuli to which any of the recorded neurons responded to, and then, in follow-up sessions, presented these response-eliciting stimuli many times (between 25 and 35 repetitions). The rationale for this number of repetitions was first, to statistically compare the neuron's responses to the different stimuli for those neurons responding to more than one stimulus, and second, to evaluate LFP responses and their relationship with the single neuron responses. Our results (from the follow-up sessions) show a strong relationship between the spike and LFP responses, with the latter consistently preceding the former, and that most MTL neurons responding to more than one stimulus exhibit "neural unitization"—i.e., they respond equally to the different stimuli eliciting significant responses, or in other words, if a neuron fires to more than one stimulus, the responses to these stimuli are indistinguishable from each other. We postulate that such "unitized" coding is the basis for encoding long-term associations in the human MTL, and is crucial to understand the mechanisms that underlie memory coding and its capacity in the human brain.

## Results

**Experimental paradigm and neural recordings.** We recorded single neuron and local field potential (LFP) activity during 21 sessions in 6 patients with pharmacologically intractable epilepsy, who were implanted with intracranial electrodes for clinical reasons. Subjects were first shown a set of approximately 100 pictures on a computer screen, 6 times each and in pseudorandom order, to determine which pictures triggered responses in the recorded neurons. Then, the pictures eliciting responses, together with other pictures forming a set of around 15 stimuli (mean: 13.9; s.d.: 4.5), were presented again in a follow up session, between 25 and 35 times each, to compare the neuronal responses

to the different pictures (Methods). The data presented here corresponds to these follow up sessions.

**Unitization of response strength.** From the 81 responsive units, 37 were "multi-responsive", i.e., they exhibited responses to more than one picture (19 units responded to 2 pictures, 5 units to 3 pictures, 6 units to 4 pictures, and 7 units to 5 or more pictures). This led to 208 "response-eliciting pairs", i.e., pairs of stimuli eliciting responses in multi-responsive units (number of pairs per unit, mean: 5.6; s.d.: 9.9). Figure 1a shows an example of a unit responding to the picture of a Boeing airplane, the interior of an airplane cabin, and the actor Leslie Nielsen (who had a major role in the 1980 movie "Airplane!").

First, we checked that the finding of multiple responses could not be attributed to spurious spike sorting. For this, we performed a permutation test—shuffling the label of the stimuli eliciting each spike—to assess whether the spikes in response to each stimulus (of the stimulus pairs) differed from each other (Methods). We found that only 5 of the 208 pairs (2.4%) had a significantly different average spike shape, thus reaffirming our assessment that multiple responses corresponded to the same neurons.

Next, we analyzed differences in the response strength between response-eliciting pairs using permutation tests (i.e., shuffling the label of the stimulus of each trial; see Methods). Figure 1b shows that the strength differences for each response-eliciting pair of Fig. 1a fall within the surrogate distributions, and therefore all three pairs showed no significant differences (stimuli 10–18, $p = 0.27$; stimuli 10–15, $p = 0.61$; stimuli 15–18, $p = 0.63$). In addition, we used a decoding approach to see if the identity of each stimulus of the multiple responses could be predicted based on the single trial response strength (Methods). Figure 1c shows the confusion matrix for the three responses shown in Fig. 1a, where the decoding performance (14%) was not significantly different from chance ($p = 0.92$).

We repeated the same analysis for all 208 response-eliciting pairs and found that only 14% of them showed a significant difference ($p < 0.05$) in the response strength (permutation test). Supplementary Fig. 1a shows the distribution of $p$-values. Regarding the decoding performance for all 37 multi-responsive units, in only 8 cases (22%) it was possible to predict (above chance) the stimuli to which the neurons responded. The distribution of the obtained $p$-values is shown in Supplementary Fig. 1b.

Figure 2a shows the response strength (baseline subtracted) for all the response-eliciting stimuli in all 37 multi-responsive units. There is in fact a wide range of responses passing the responsiveness criterion, but note that most neurons show responses that tend to cluster together (e.g., the exemplary units presented in Fig. 1a and Supplementary Fig. 7), whereas a few exhibit large differences (e.g., Supplementary Fig. 8). To further quantify the similarity between the responses of the "multi-responsive" neurons, we compared the differences between pairs of responses in the same neuron with pairs of responses from different neurons. For each responsive pair, we constructed 1000 surrogate pairs by mixing responses from different neurons (Methods) and found that the median of the response-eliciting pair differences was smaller than the medians of the surrogate distributions (i.e., $p < 10^{-3}$). Figure 2b shows the original and a representative surrogate distribution of strength difference (the one with the median difference across all surrogate distributions), which were significantly different (rank-sum, $p \sim 10^{-30}$).

Given that a lack of significant differences in the response strength could be due to limited statistical power (i.e., having relatively few trials), we computed the normalized response strength difference for all response-eliciting pairs as a function of the number of trials used in the estimation (Methods).

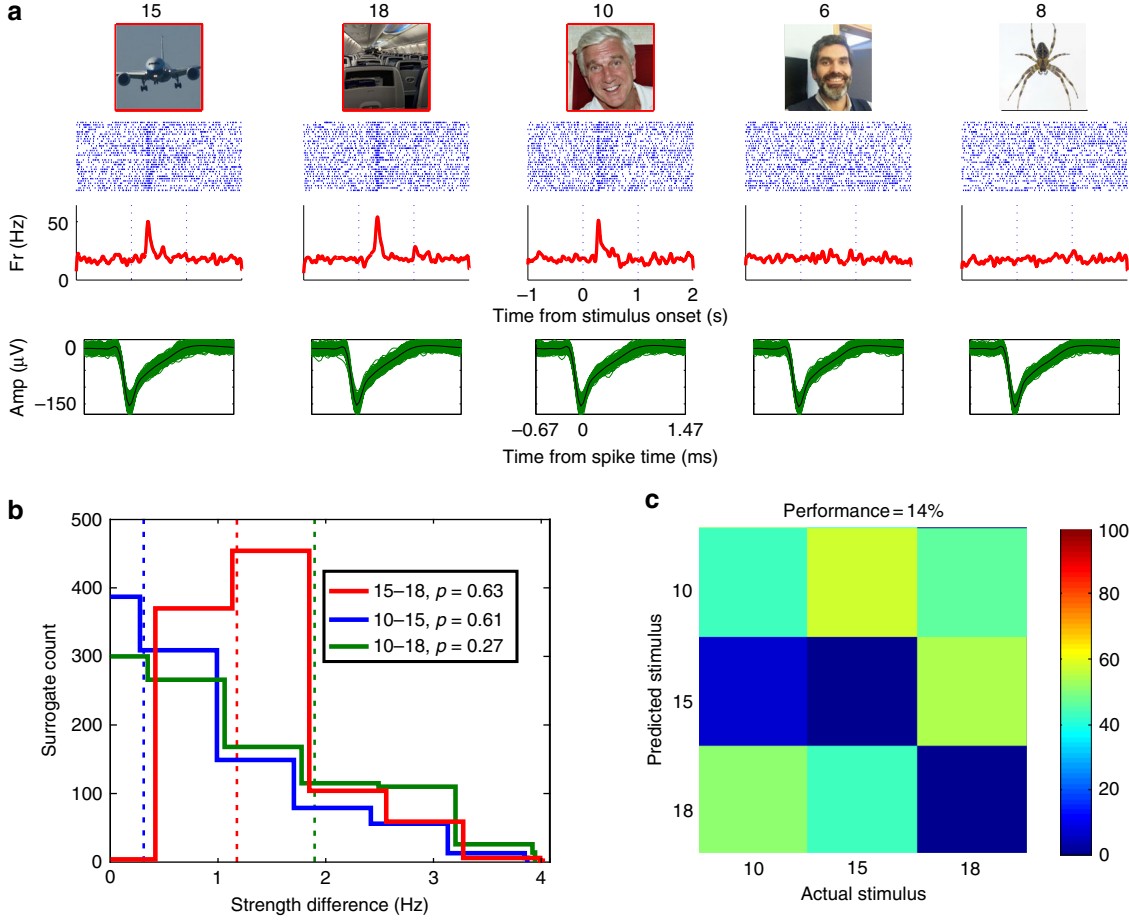

**Fig. 1** Exemplary multi-responsive unit. **a** Responses of a unit in the left hippocampus. For each stimulus, the raster plot (blue lines represent the appearance of a spike and each row is associated to a trial; first trial is at the top and time zero is the stimulus onset), instantaneous firing rate, and spike shapes in the response period (between 100 and 800 ms after stimulus onset), are shown. Stimulus numbers appear at the top of the stimulus pictures. The unit responded to the picture of a Boeing, the interior of a cabin airplane, and the actor Leslie Nielsen (who had a major role in the 1980 movie "Airplane!"). The association scores for this unit (Methods) were $AS_{R-R} = 0.3$ and $ASR_{N-R} = -0.06$, indicating a strong association between the response-eliciting stimuli. Note that the spike shapes are the same for all responses, indicating that they come for the same neuron. **b** Surrogate distributions of strength difference for each response-eliciting pair in this unit. Vertical dashed lines represent the actual strength difference for each pair. All three pairs showed no significant difference (stimuli 10–18, $p = 0.27$; stimuli 10–15, $p = 0.61$; stimuli 15–18, $p = 0.63$). **c** Confusion matrix based on the single trial spike count of the response-eliciting stimuli (the color code represents the percentage of trials where stimulus $i$ was presented and the decoder labeled it as stimulus $j$). Stimulus numbers are the same as in **a**. The decoding performance was 14%, not significantly different from chance ($p = 0.92$). Due to copyright issues, the images presented here are similar to the ones actually presented to the subjects. Copyright notes: Picture 6 is a self-portrait from Dr Antonio Valentin (co-author of the paper). Picture 10 was cropped from "Leslie Nielsen" by Alan Light, licensed under CC BY 2.0. Picture 8 was cropped from "Spider2007-09-03" by Trounce, licensed under CC BY-SA 3.0. Picture 18 was cropped from "Airplane Cabin 1 2017-06-18" by FASTILY, licensed under CC BY-SA 4.0. Picture 15 was cropped from "Front view of B787 Approaching at Oshkosh 2011" by H. Michael Miley, licensed under CC BY-SA 2.0

Supplementary Fig. 2 shows that the estimation of strength differences became stable when using 18 trials or more (less than 1% change), which is far less than the 29 trials we used on average (with a minimum of 25).

Next, we quantified the normalized strength of the activity of every responsive unit to both response- and non-response-eliciting stimuli (Methods). As can be seen in Fig. 2c, a neuron shows zero strength in response to most of the non-response-eliciting stimuli, whereas it responds similarly and with its maximum strength to most of the response-eliciting stimuli. These results point towards a nearly binary code, with neurons mainly responding to the stimuli with maximum (minimum) strength, without a typical Gaussian-like tuning peaking at the maximum response.

In addition, we compared the spike count during baseline and response periods, both for the individual responses and for the

multiple responses of the same neurons (Methods). Supplementary Fig. 3 shows that, when comparing mean, variability and coefficient of variation of the spike count (strength), there were significant differences between baseline and the individual responses, but not between individual and multiple responses.

Altogether, these results show that different responses in multi-responsive neurons have, in general, no significant differences in strength.

**Unitization of single neuron and LFP response latencies.** Next, we looked for latency differences in both the spike and LFP responses. Similarly to the analysis performed previously with the strength difference, to verify that our results were not biased due to the number of trials used, we computed the latency of the spike responses as a function of the number of trials (Methods). The

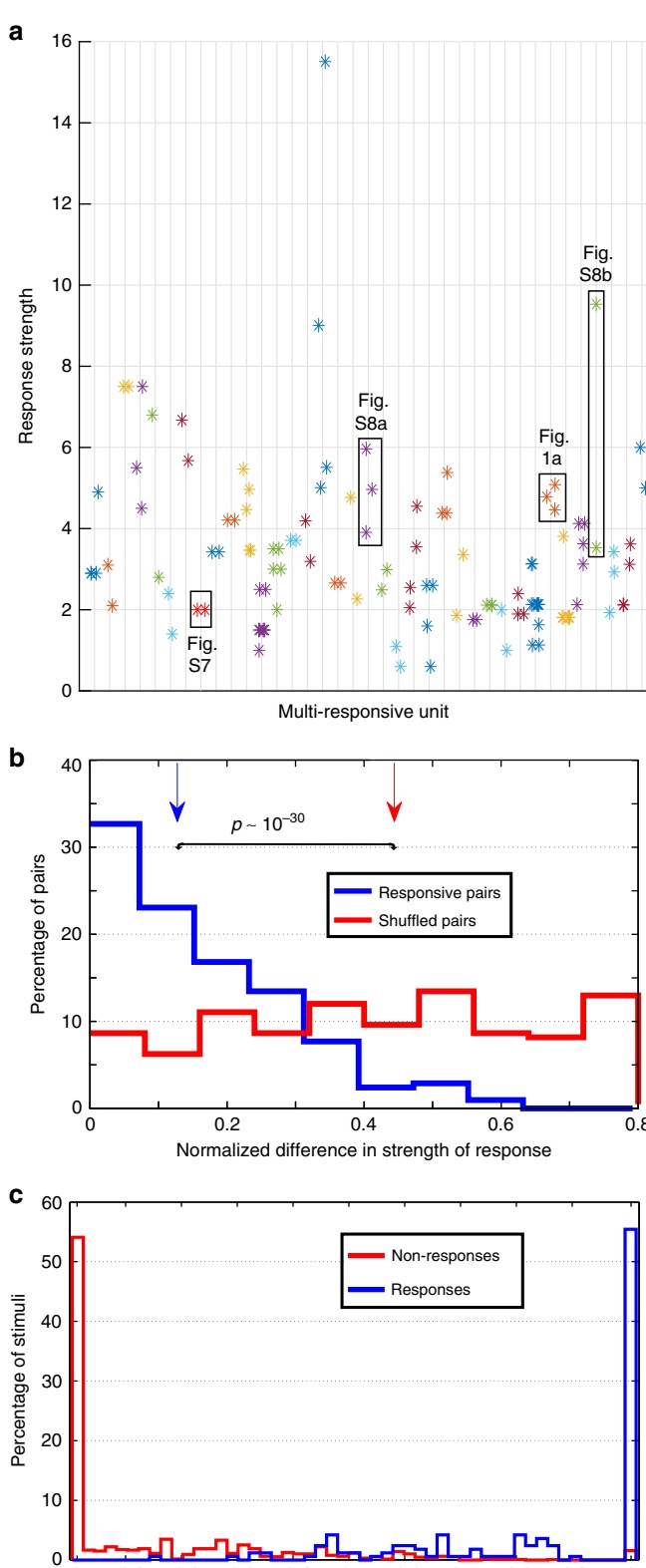

**Fig. 2** Multi-responsive units do not exhibit differences in response strength. **a** Strength (baseline corrected) for all the response-eliciting stimuli in all 37 multi-responsive units. Some of the exemplary responses shown in this work are highlighted. **b** Distribution of normalized strength difference for the response-eliciting pairs and for a representative case of randomly chosen pairs (Methods). The difference in the response-eliciting pairs was significantly smaller than the one on a representative distribution of shuffled pairs ($n = 208$, one-sided rank-sum test, $p \sim 10^{-30}$). Vertical arrows denote the median of the distributions. **c** Histograms for the normalized strength of activity in all the responsive units, computed for all the stimuli presented in each session, and separated according to whether or not they were responsive

Strong average LFP theta power responses have been previously reported following picture presentation[24]. Therefore, in the current study we focused on LFP responses in the theta band. Figure 3a shows an exemplary response (to the picture of Stonehenge) from the left hippocampus. The presentation of the picture of Stonehenge led to a clear increase in the evoked theta power (black trace), from which we could estimate a response latency onset (black dashed line) (Methods). The latency of the evoked LFP response (103 ms) was earlier than the one of the spike response (221 ms). Since different neurons show different spike response latencies (range: 111–653 ms; mean: 287 ms), we performed a latency corrected grand average, where the LFP power traces of the individual responses were aligned to the spike latency before computing the average. As shown in Fig. 3b, the LFP response appeared, on average, 50–100 ms earlier than the spike response. Furthermore, Fig. 3c shows that there was a significant correlation between the LFP and spike response latencies (Pearson correlation, $r = 0.35$, $p = 4 \times 10^{-3}$). The distribution of latencies projected orthogonally to the axis $y = x$ had a clear peak, corresponding to the LFP latency appearing ~70 ms earlier than the spike latency. To further show this effect, we split the response set in two subsets based on the median of the spike latency (278 ms), and computed the grand averages for each subset (early vs. late responses). Figure 3d shows that the instantaneous firing rates for the early and late spike responses were separated by construction, but their corresponding LFP traces were also separated (peak latency early vs. late, rank-sum, $p = 3 \times 10^{-3}$). As before, the LFP trace of each group increased before the corresponding instantaneous firing rate.

Figure 3c, d show that spike and LFP latencies cover a wide range of values for individual responses. This range could be due to different neurons having different latencies, or it could be that different stimuli are associated with different latencies even in the same neuron. To address this, we focused on multi-responses and compared the latencies of the response-eliciting pairs. We performed permutation tests on the spike and LFP response latencies in the response-eliciting pairs (Methods) and found that the percentage with significant differences ($p < 0.05$) was 19% and 4%, respectively. Supplementary Fig. 1c, d show the distribution of p-values. Furthermore, following the same approach used in Fig. 2a for the strength analysis, we found that the median difference in spike latency for the original response-eliciting pairs was smaller than the distribution of medians from the surrogate distributions (i.e., $p < 10^{-3}$). In addition, Fig. 4a shows that the difference in spike latency in the population of response-eliciting pairs was significantly smaller than that of a representative distribution of surrogate pairs (rank-sum, $p \sim 10^{-10}$). Similarly, the median difference in the LFP latency for the original response-eliciting pairs was significantly smaller than the distribution of medians from the surrogate distributions ($p < 10^{-3}$), and the response-eliciting pairs showed significantly

top panel of Supplementary Fig. 4 shows that the estimate became stable when using more than 11 trials. Moreover, the bottom panel shows that differences when considering more trials were not due to the order in the trial sequence (e.g., due to habituation effects). In fact, the estimation based on trials 1 to 6 was not significantly different than the one based on trials 20 to 25 (paired sign test, $p = 0.93$).

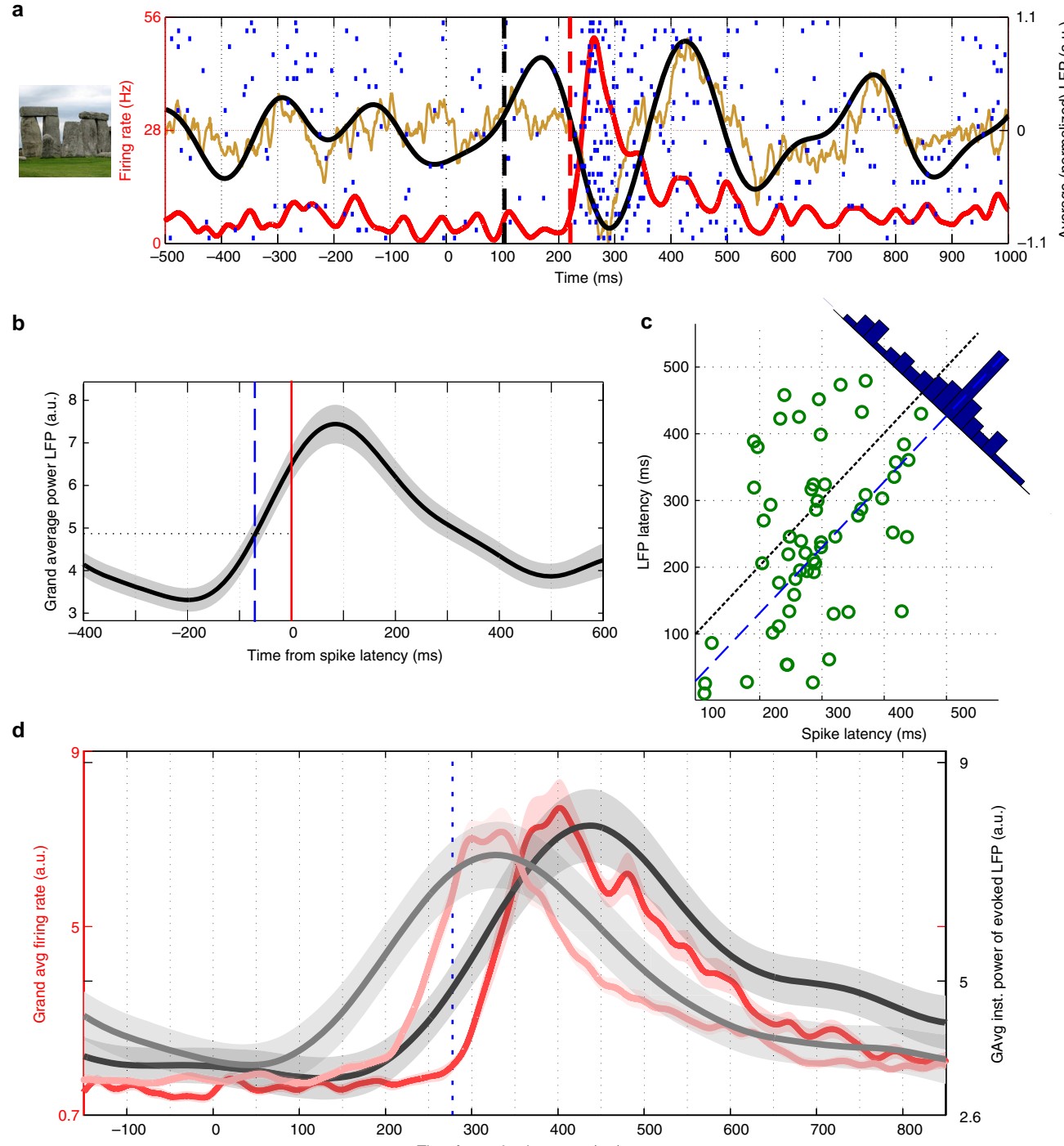

**Fig. 3** LFP and spike latency analysis for individual responses. **a** Exemplary unit recorded in the left hippocampus that responded to the picture of Stonehenge. The red curve corresponds to the instantaneous firing rate, whereas the vertical dashed line marks the spike response onset. The average LFP responses are shown in brown (raw LFP: 2 to 512 Hz) and black (theta LFP: 3 to 6 Hz), with the vertical dashed line marking the LFP response onset (Methods). The spike response latency was 221 ms and the LFP onset occurred 118 ms before. Due to copyright issues, the image presented here (cropped from "Stonehenge 02" by Bernard Gagnon, licensed under CC BY-SA 3.0) is similar to the one actually presented to the subject. **b** Latency corrected average of the 151 theta LFP responses. The LFP traces of the individual responses are aligned to the spike latency before computing the grand average. The blue dashed line marks 70 ms before the spike latency. **c** LFP and spike latencies for individual responses were significantly correlated (Pearson correlation, $r = 0.35$, $p = 4 \times 10^{-3}$). Points were projected into the orthogonal direction to $y = x$ (black dashed line) and their distribution shows a clear peak. The blue dashed line represents $y = x - 70$. **d** Grand average of the instantaneous firing rate (red) and the theta LFP (black) for the "early" (light traces) and "late" (dark traces) subsets of responses, which were split with respect to the median spike latency (278 ms, blue dashed line). The peak LFP latency was significantly different for the early and late groups (one-sided rank-sum test, $p = 3 \times 10^{-3}$) and for both groups the LFP responses preceded the ones of the spikes

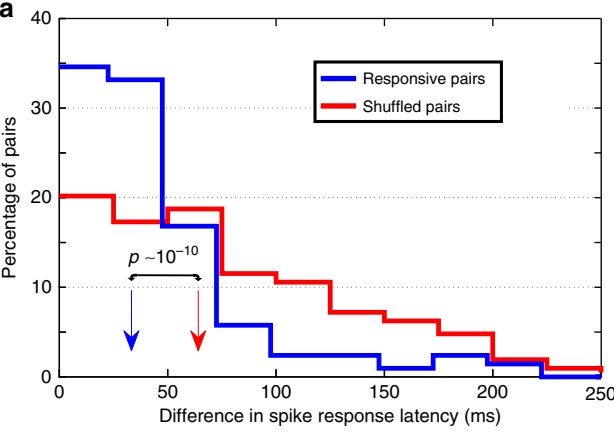

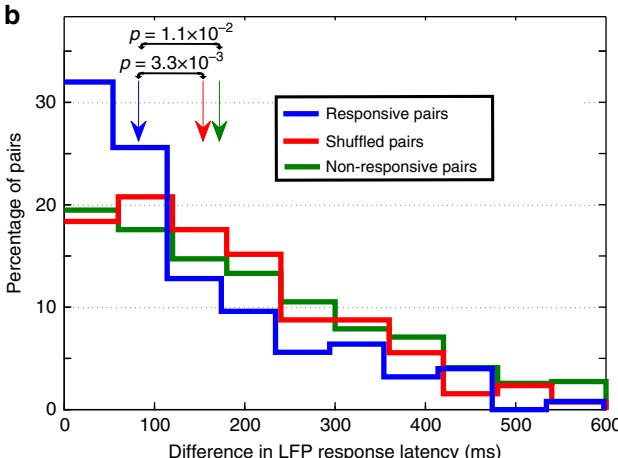

**Fig. 4** Multi-responsive units do not exhibit differences in spike or LFP response latencies. **a** Distribution of spike latency difference for the response-eliciting pairs and for a representative case of randomly chosen pairs. The arrows mark the median of the distributions, which were significantly different ($n = 208$, one-sided rank-sum test, $p \sim 10^{-10}$). **b** Same as **a** but for the LFP response latency difference. The response-eliciting pairs showed significantly less difference in LFP latency than the one obtained on a representative distribution of shuffled pairs ($n = 125$, one-sided rank-sum test, $p = 3.3 \times 10^{-3}$). In addition, the distribution of latency differences for the response-eliciting pairs was significantly smaller than for non-response-eliciting pairs ($n = 6839$ for non-response-eliciting pairs, one-sided rank-sum test, $p = 0.011$)

less difference in LFP latency than the one obtained from a representative distribution of surrogate pairs (Fig. 4b, rank-sum, $p = 3.3 \times 10^{-3}$). Since we have previously observed that most "non-response-eliciting stimuli"—i.e., those that do not elicit a spike response in a neuron recorded with a certain electrode—do however elicit an LFP response[24], we studied the latency difference for non-response-eliciting pairs. We observed that the distribution of latency differences for the response-eliciting pairs was significantly smaller than for non-response-eliciting pairs (Fig. 4b, rank-sum, $p = 0.011$), showing that the similarity of LFP latencies is a property of response-eliciting pairs, likely due to the fact that these stimuli tend to be associated (see below). These results show that, in general, and as was the case for the response strength, the response latencies do not discriminate between the different response-eliciting stimuli in a multi-responsive unit.

**Unitization in phase locking between spikes and LFPs.** To provide further insights into the relationship between the spike

and LFP responses, a phase locking analysis was performed by evaluating if the spikes appear at a particular preferred phase of the LFP in the theta band (Methods). Figure 5a shows an exemplary unit recorded in the left hippocampus exhibiting a response to the picture of Alastair Cook, former captain of the English cricket team. Spikes during the baseline period were not significantly locked to the (theta) phase of the LFP (phase locking index, PLI = 0.05, $p = 0.82$). In contrast, those in the response period showed a significant locking (PLI = 0.44, $p = 1.6 \times 10^{-3}$). At the population level, Supplementary Fig. 5 shows that this behavior can be seen in most neurons.

We further studied whether such phase locking is present at the single trial level or whether it is only an effect observed in the averaged responses. For each response with significant phase locking we computed its PLI (original), and generated surrogates by repeating this procedure after shuffling the trial labels (i.e., using different traces of the instantaneous phase to compute the PLI). Figure 5b illustrates the relationship between original PLIs and the mean of the corresponding surrogate distributions. The shuffled values were significantly smaller than the original ones (paired sign test, $p \sim 10^{-11}$), showing that the relation between spike times and instantaneous phases is present at the single trial level.

Following this, we focused again on the multi-responsive units. The wide range of preferred phases observed in Supplementary Fig. 5b could be due to different preferred phases for different neurons, or it could also be present in the different responses of the same neuron. To assess this, we compared the phase preference of multi-responsive units, using a set of 44 response-eliciting pairs with both responses being significantly phase locked. Using the same approach as in Figs. 2a, 4, we found that the median difference in the preferred phase for the original response-eliciting pairs was significantly smaller than the distribution of medians from surrogate distributions ($p = 5 \times 10^{-3}$), and that the response-eliciting pairs showed significantly less difference in preferred phase than the one obtained on a representative distribution of surrogate pairs (Fig. 5c, rank-sum, $p \sim 10^{-4}$). These results show that multi-responses are phase locked to similar preferred phases.

**Unitization as a mechanism to encode associations in the MTL.** Next, we used an association metric introduced in De Falco et al.[22] to test whether the response-eliciting stimuli in a particular neuron tended to be associated to each other. For each pair of stimuli, we defined an association score based on the number of hits found using a web search engine (Methods). In line with the results from De Falco et al., the web association score was a good proxy to evaluate the personal associations of the subjects (Methods), as there was a significant correlation between the metrics (Supplementary Fig. 6a, Spearman correlation, $\rho = 0.24$, $p = 5.9 \times 10^{-3}$). For each multi-responsive unit, the normalized scores were averaged within two groups depending on whether both stimuli led to significant responses (R–R), or one elicited a response and the other did not (R–NR). Therefore, for each multi-responsive unit, we calculated a mean association score for pairs of responses ($AS_{R-R}$) and for the other pairs ($AS_{R-NR}$). Figure 6a shows that the association scores for R–R were significantly larger than for R–NR (paired sign test, $p = 0.02$), thus showing that if a neuron is responsive to more than one stimulus, these stimuli tend to be associated. It should be noted that nearly all the associations we studied here have already been developed long before running the experimental paradigm, since passive viewing of 30 presentations of about 15 stimuli in pseudorandom order does not explicitly promote the development of new associations.

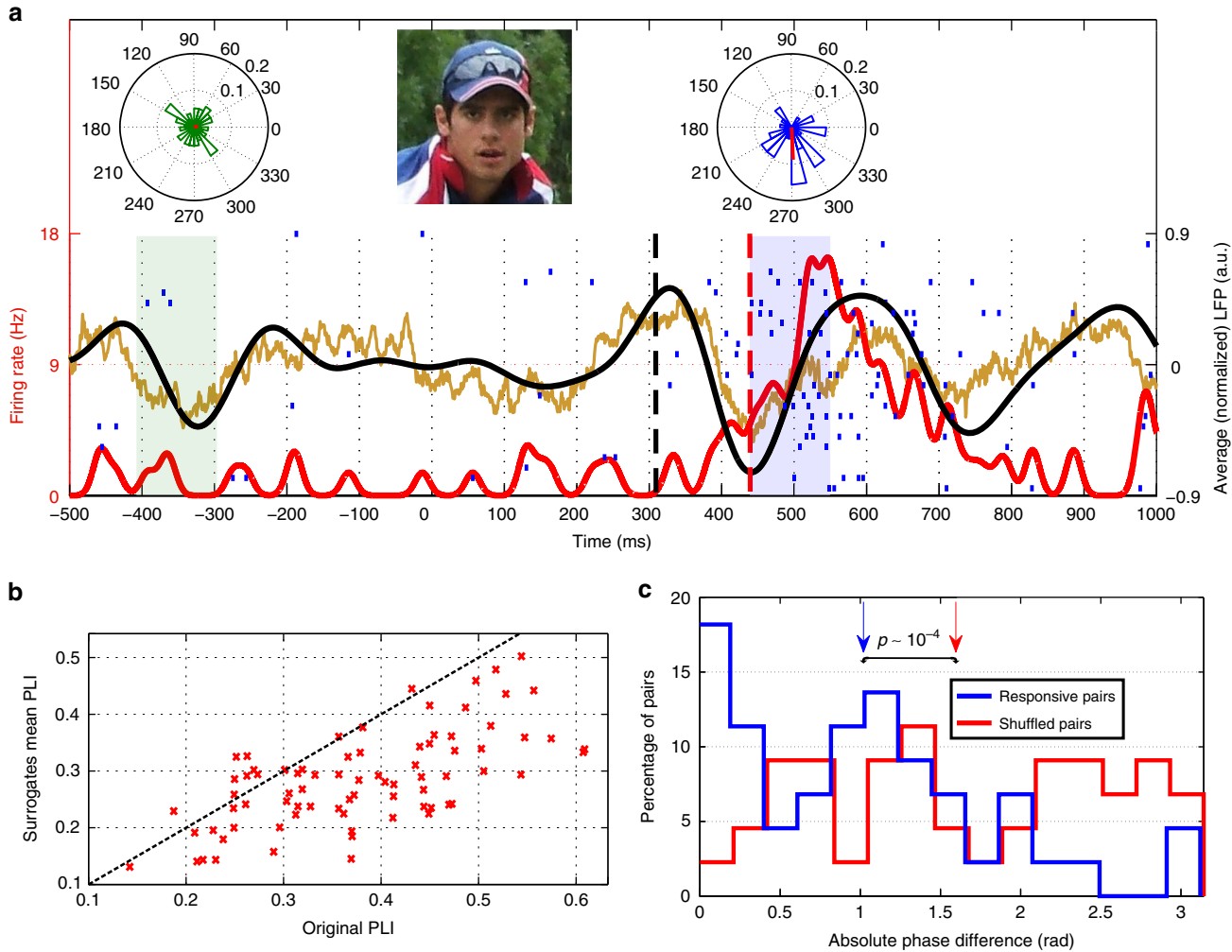

**Fig. 5** Phase locking analysis for individual and multiple responses. **a** Exemplary unit recorded in the left hippocampus exhibiting a response to the picture of Alastair Cook, former captain of the English cricket team. Same conventions as in Fig. 3a. The spike response latency is 439 ms, with the LFP onset occurring 130 ms earlier. Circular histograms of the phase at the spike times during baseline (green, left) and response (blue, right) periods (marked with green and blue shaded areas, respectively). Red lines represent the mean phase vector. Spikes during the baseline period are not significantly phase locked (PLI = 0.05, $p$ = 0.82), in contrast to those in the response period (PLI = 0.44, $p$ = 1.6 × 10$^{-3}$). Due to copyright issues, the image presented here (cropped from "Alastair cook bowl" by BInguyen, licensed under CC BY-SA 3.0) is similar to the one actually presented to the subject. **b** PLI for the significant phase-locked responses (actual vs. mean of the surrogate distribution). The mean shuffled values were significantly smaller than the original ones ($n$ = 82, one-sided paired sign test, $p$ ~ 10$^{-11}$), showing a relation between the spike times and the instantaneous LFP phases at the single trial level. **c** Distribution of mean phase difference for the responsive and surrogate pairs. The difference in the response-eliciting pairs was significantly smaller than the one on a representative distribution of surrogate pairs ($n$ = 44, one-sided rank-sum test, $p$ ~ 10$^{-4}$). Vertical arrows denote the median of the distributions

We explored some potential alternatives to the association between stimuli being the relation among multiresponses. First, we explored whereas results could be explained in terms of broad semantic category responses. For this, we classified the individual stimuli into 8 categories and compared the associations scores for all pairs of stimuli belonging to the same category pairs (Methods). We found a significant difference between $AS_{R-R}$ and $AS_{R-NR}$ (Supplementary Fig. 6b, paired sign test, $p$ = 0.04). Therefore, the results cannot be explained by semantic categorization. We also ruled out the possibility of stimulus familiarity and visual similarity (Methods), as we found no significant differences for these scores between R–R and R–NR (familiarity, Supplementary Fig. 6c, paired sign test, $p$ = 0.63; visual similarity, Supplementary Fig. 6d, paired sign test, $p$ = 0.74).

By defining the high association group as the set of pairs with an association score above its first quartile (i.e., considering 75% of the pairs with the largest association scores), we found that for

this group: 1) the percentage of pairs with no significant strength difference went up from an overall 86% to 90%; 2) the percentage of pairs with no significant spike latency difference went up from an overall 81% to 87%; 3) the percentage of pairs with no significant difference in both spike strength and latency went up from an overall 71% to 78%. This suggests that when the stimuli are associated, the neural responses tend to be more unitized. To illustrate this result, note that the stimuli to which the unit in Fig. 1a responded to were highly associated ($AS_{R-R}$ = 0.3 and $AS_{R-NR}$ = −0.06) and exhibited no differences in response strength. The same applies for the exemplary response in Supplementary Fig. 7, where $AS_{R-R}$ = 1.8 and $AS_{R-NR}$ = 0.2). On the contrary, Supplementary Fig. 8 shows two exemplary units that exhibited significant differences in response strength following the decoding analysis, but the stimuli eliciting responses in these units were not associated (Supplementary Fig. 8a, $AS_{R-R}$ = −2.5 and $AS_{R-NR}$ = −0.5; Supplementary Fig. 8b,

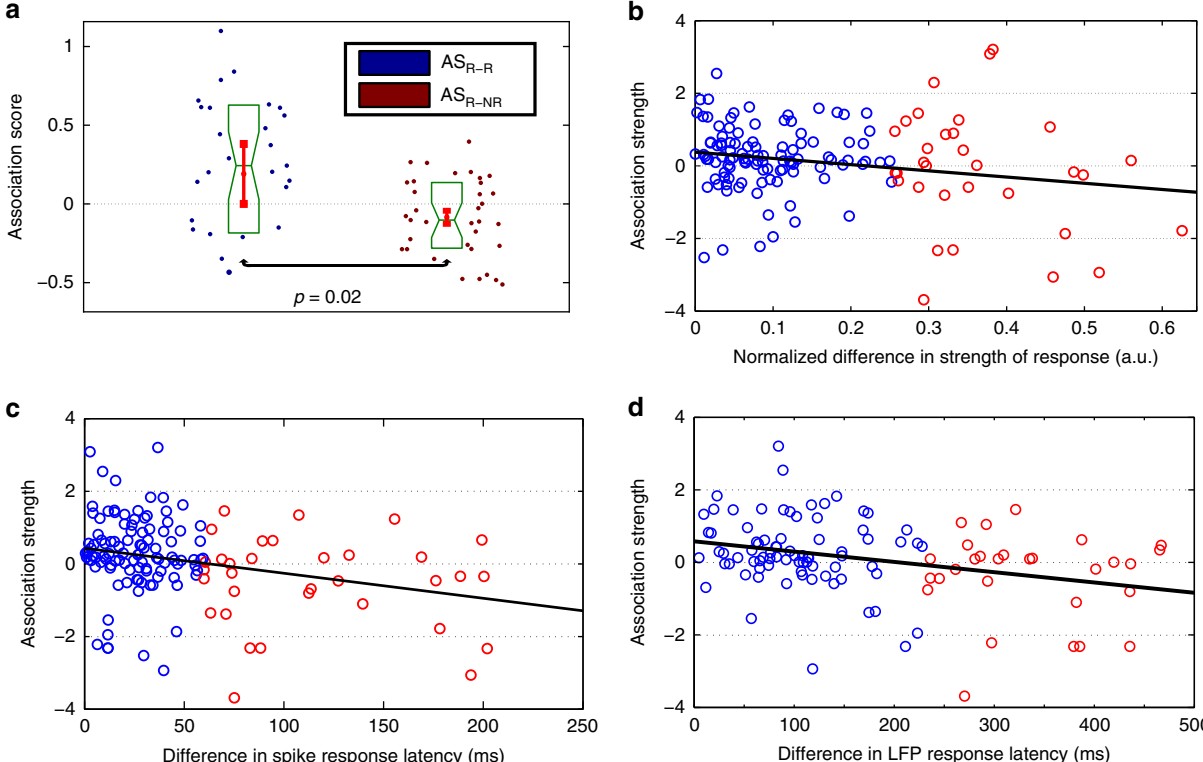

**Fig. 6** Neural responses are more similar when the stimuli are associated. **a** Web-based mean association scores for multi-responsive units on stimulus pairs where both stimuli were responsive ($AS_{R-R}$), or one was responsive and the other one not ($AS_{R-NR}$). Response-eliciting pairs have a significantly larger association score ($n = 35$, one-sided paired sign test, $p = 0.02$). Each dot represents a multi-responsive unit. Mean ± standard error of the mean are shown in red, whereas boxplots are in green (center line, median; box limits, upper and lower quartiles; notch limits, (1.57 × interquartile range)/sqrt($n$)). **b** Association score as a function of the normalized difference in the response strength. Both quantities were significantly correlated ($n = 139$, Pearson correlation, $r = -0.2$, $p = 0.019$). The pairs representing the 25% with the largest differences are shown in red. **c** Same as **b** for the spike latency difference ($n = 139$, Pearson correlation, $r = -0.28$, $p \sim 10^{-4}$). **d** Same as **c** for the LFP latency difference ($n = 93$, Pearson correlation, $r = -0.22$, $p = 0.014$)

$AS_{R-R} = -2.9$ and $AS_{R-NR} = -0.69$). Altogether, in 6 out of 8 units showing a significant decoding performance (i.e., differences in the response strength) we observed that the stimuli the neuron fired to were not associated, i.e., $AS_{R-R} < AS_{R-NR}$.

To further quantify the role of stimulus association in the similarity of electrophysiological responses, we studied the correlation between the association score for the individual response-eliciting pairs and different electrophysiological measures (Methods). There was a significant correlation between the association score and the normalized difference in the response strength (Fig. 6b, $r = -0.2$, $p = 1.9 \times 10^{-2}$), spike (Fig. 6c, $r = -0.28$, $p \sim 10^{-4}$), and LFP latency differences (Fig. 6d, $r = -0.22$, $p = 1.4 \times 10^{-2}$). We also explored if these correlations were present for the R–NR pairs when looking at the difference in strength and LFP latency (not for spike latency as most non-response-eliciting stimuli did not elicit a neural response from which a spike latency could be defined). There was no correlation between the association strength and the normalized strength difference (Supplementary Fig. 9a, $r = 0.01$, $p = 0.74$), with significant differences when compared with the correlation for R–R pairs (Fisher Z transformation, $p = 9.2 \times 10^{-3}$). The same was the case for the LFP latency, with no correlation for R–NR (Supplementary Fig. 9b, $r = -0.05$, $p = 0.08$), while being significantly different to the one for the R–R pairs (Fisher Z transformation, $p = 1.5 \times 10^{-2}$).

Furthermore, we split the sets based on the third quartile of normalized strength, spike and LFP latency (the groups representing the 25% with the largest differences are shown in

red in Fig. 6b–d). By focusing on the groups with the smaller differences (each representing 75% of the pairs), the resulting correlations vanished (strength: $r = 0.03$, $p = 0.77$; spike latency: $r = -0.05$, $p = 0.63$) or were reduced (LFP latency: $r = -0.15$, $p = 0.15$) with respect to the ones observed in the whole sets. Moreover, a comparison of the correlation for these subsets and the ones associated for R–NR pairs showed no significant differences for either strength (Fisher Z transformation, $p = 0.85$) or LFP latency (Fisher Z transformation, $p = 0.23$).

Overall, these results suggest a mechanism of "neural unitization" for encoding long-term associations in the human MTL, whereas the observed global correlations in Fig. 6 are driven by the minority of non-unitized and non-associated pairs. In this context, the response to each of the associated stimuli is unitized, meaning that they show similarity in terms of response strength, spike and LFP response latency, and preferred phase for locking between spikes and LFPs, i.e., they have a similar "electrophysiological signature".

**A putative mechanism for neural unitization.** Spike latencies of individual responses span a wide range of values (Fig. 3c). Yet, we found that multiple responses in individual neurons have similar strength and latency. If we have two cell assemblies in the MTL, each one encoding a particular stimulus, the presentation of each stimulus will lead to the activation of the corresponding assembly, with a certain latency, through an ignition process, i.e., a fast-nonlinear activation[25–28]. When these two stimuli become strongly associated, some neurons will start firing to both stimuli

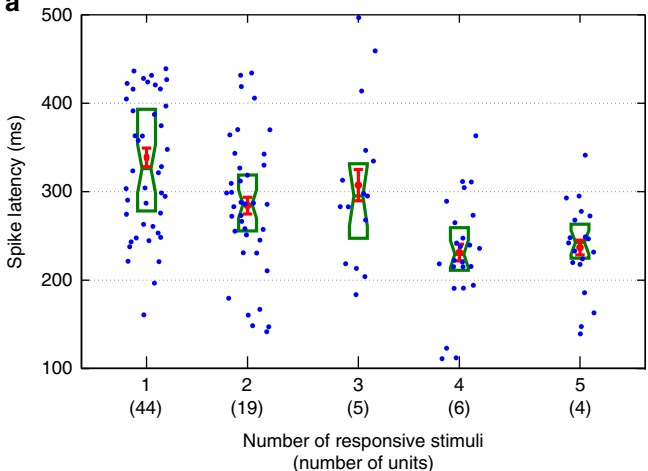

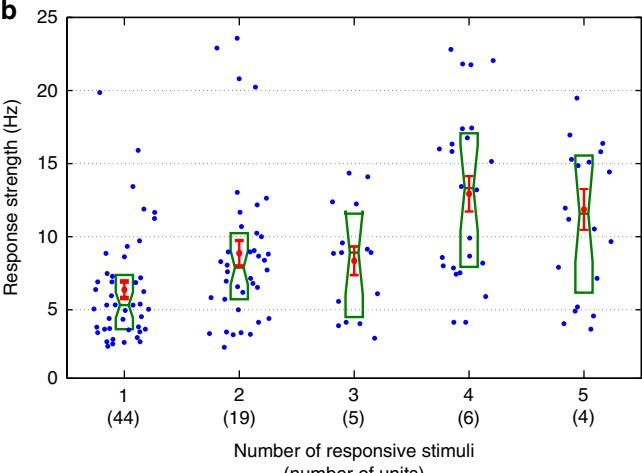

**Fig. 7** Neural unitization as a mechanism for encoding associations in the human MTL. **a** Spike response latency was significantly correlated with the number of response-eliciting stimuli (Spearman correlation, $\rho = -0.45$, $p \sim 10^{-8}$). Each dot represents an individual response. Mean ± standard error of the mean are shown in red, whereas boxplots are in green (center line, median; box limits, upper and lower quartiles; notch limits, (1.57 × interquartile range)/sqrt($n$)). **b** Same as **a** but showing that multi-responsive neurons have a larger response strength (Spearman correlation, $\rho = 0.43$, $p \sim 10^{-7}$)

and will then encode the association by unitizing the neural responses. In this context, we hypothesized that the unitized response will show the latency associated to the earliest latency of each assembly, i.e., the one starting the ignition process. This leads to the prediction that the spike response latency should be smaller for units firing to more stimuli (as there are more stimuli that could shorten the latency of the responses). Figure 7a shows the relationship between the spike response latency and the number of response-eliciting stimuli in each unit (Methods). In line with our prediction, and in spite of the fact that we do not have access to all the responses from the neurons, we found smaller latencies for neurons responding to more items ($\rho = -0.45$, $p \sim 10^{-8}$). In addition, we found that multi-responsive neurons have a larger response strength (Fig. 7b, $\rho = 0.43$, $p \sim 10^{-7}$), and the same result was obtained for the baseline strength. We also noticed that the baseline activity was negatively correlated with the spike latency for each multi-responsive neuron ($r = -0.27$, $p = 2.6 \times 10^{-3}$). To rule out that

the effect in Fig. 7a was due to a bias in the latency estimation (as neurons with lower baseline activity will tend to give less accurate and higher latency estimations), we chose 20 units from both the single and multi-responsive groups with matching baseline strengths and found significantly earlier latencies for the multi-responsive group (one-sided rank-sum test, $p = 8.7 \times 10^{-3}$).

## Discussion

The concept of "unitization" has been largely used in the Psychology literature[17,29–32]. In this sense, unitization involves representing previously separate items as a single entity. Here, we use the concept of "neural unitization" to refer to something different, namely whether different stimuli can be discriminated based on their neural responses, but without implying whether such stimuli can be discriminated or not at the behavioral level.

We found that most MTL multi-responsive neurons exhibited no difference in the strength and latency of their responses to the different stimuli. This was particularly the case for stimuli that were largely associated to each other. We also found a correlation between spike and (theta) LFP latencies, with the latter preceding the former by approximately 70 ms. In addition, we showed that the timing of the neuron's response onset is not solely given by the time of stimulus presentation, but that there is also a fine tuning at the single trial level according to the instantaneous LFP phase. As for the single neuron responses, most multi-responsive units also showed no difference in LFP latency across response-eliciting stimuli, nor in the preferred phase of locking of the spikes to the LFP phase. Furthermore, the LFP carried information about the stimulus, since associated stimuli had more similar latencies than non-associated ones.

In contrast to the unitized responses we observed for most human MTL neurons, neurons in cortical areas typically exhibit tuning curves, with one or a few preferred stimuli eliciting strong responses, and the other stimuli showing weaker graded responses according to their degree of similarity with the preferred one/s in a certain feature space. Among others, graded neuronal responses have been observed in the cat primary visual cortex[33], the cercal system of the cricket[34], the rat auditory[35], visual[36] and somatosensory[37] cortices, the dorsal processing stream in monkeys, including the posterior parietal cortex[38] and M1[39], as well as the ventral visual processing stream, including V1[40], V4[41], the inferotemporal cortex (IT)[42–44], and the face patches[45,46].

Many of the studies describing graded responses in animals used similar experimental paradigms to ours, i.e., passive viewing of images. Specifically, cells in the monkey IT cortex show sparse responses to specific faces[46], with its responses being (so far) the closest to the responses we have described in the human MTL[47]. However, Chang & Tsao[48] recently showed that these neurons also display a graded representation, and respond according to the projection of the faces along particular axes in a high-dimensional space describing shape and appearance of the faces. Other studies have also shown categorical responses in IT but without neural unitization, since from the tuning of these neurons it was possible to discriminate the identity of the items within a category[49,50].

Since IT neurons have a large number of direct projections to the MTL[51], it is interesting to have found a unitization mechanism for encoding long-term associations in the MTL.

Evidence from studies in animals[1,9–16] and humans[4,17–20], have emphasized the importance of the MTL in the encoding of associations. Paradigms using items and reward location associations have been used in monkeys while hippocampal neurons were recorded[7,10,52]. However, the design of these studies did not allow the examination of the unitization of the neural responses,

since the responses to the individual locations (i.e., without the item) could not be assessed on their own and eventually be compared to the responses to the individual items. A notable exception is the work by Fujimichi et al.[16], which used a pair association task in monkeys and found unitization of the response strength in area 35 of perirhinal cortex (but not in area 36, which is the one typically targeted for recordings). Here, we compared not only the strength, but also the spike and LFP latencies and the relationship between spike timing and LFP phase. Moreover, in those paradigms with monkeys, the animals associate arbitrary meaningless stimuli (such as fractals) after extensive training, as opposed to the long-term encoding of meaningful associations we studied here, which was not enforced by an explicit associative learning task.

It has been recently shown in Ison et al.[21] that when humans learn new associations, neurons in the MTL exhibit a graded firing that was enough to discriminate between the item originally coded by the neuron and the one associated with it. In fact, in 38% of the cases there was a spike latency difference, and in 71% a strength difference between the item originally encoded and the associated one. In contrast, in this work we dealt with long-term associations during passive viewing, which were naturally formed from the subject's own experiences before the experiments took place. For the highly associated stimuli, we observed a latency difference in only 13% of the cases, and a strength difference in only 10% of the cases. Furthermore, there was no difference in response strength and latency in only 19% of the cases reported in Ison et al., whereas for the long-term associations studied here, we found that in 78% of the cases. These evidence supports the idea that neural unitization constitutes a neural mechanism underlying the encoding of long-term associations in the MTL.

We also found a significant correlation between the strength of the spiking activity and the number of response-eliciting stimuli (Fig. 7b), which could be the result of the neurons being strongly wired into the network, thus receiving more excitatory drive[53,54], and/or that they have a higher intrinsic excitability and, consequently, a higher chance of being recruited to encode associations[55]. However, the data from Ison et al. points towards the first option, as no significant difference in baseline activity was found between the neurons that successfully encoded associations and the ones that did not. We also found that the more stimuli a unit responds to, the earlier its response latency. We propose that this is the result of the process of neural unitization: the presentation of a given stimulus ignites the firing of a cell assembly in the hippocampus and, if a unit has been recruited to encode associations, it will respond equally to all the associated items and will adopt the earliest latency among the ones of the individual assemblies, i.e., the latency of the stimulus that first starts such an ignition process.

It has been postulated that neurons in the human MTL represent the meaning of the stimulus for declarative, and particularly episodic, memory functions, and that each item is encoded in an assembly of "concept cells" that, when activated, brings the specific concept into awareness[6]. Supporting this view, the great majority of the responses to associated stimuli were unitized, thus arguing against the notion of representations at the single neuron level popularly known as "grandmother cell"[56,57] coding. In fact, by encoding associations, MTL neurons lose the identity of the concept they initially responded to, and the information about the concept identity remains encoded at the neural assembly level.

The mechanism of neural unitization described here provides a simple and flexible way of encoding associations in the MTL, and therefore memories, which contrasts with the graded responses observed in cortex. Memories might be also stored in cortex after consolidation, and a graded representation may be present there

as well. However, the MTL, and particularly the hippocampus, provides an exquisite machinery to rapidly associate any arbitrary stimuli[58–60], which is a key feature of episodic memory. When such associations are stored in long-term, neurons in the MTL encode them by unitizing the neural responses. Furthermore, unitized responses may offer important advantages for memory coding. Specifically, it has been shown that a graded code can potentially store more information per pattern, but it reduces the efficacy for retrieval due to interference between patterns[61]. In line with this, the unitized code presented here, together with the high sparseness of these neuron's responses[6], increases the effective capacity for memory storage and successful retrieval. Notably, these results impose critical constraints in the development of theoretical models of memory function and capacity[5,58–60], shedding light onto how memories are encoded in the MTL.

## Methods

**Subjects and recordings**. We report results from 21 experimental sessions in 6 patients with pharmacologically intractable epilepsy (all right-handed, four males, 23–56 years old). Patients were implanted with chronic depth electrodes at King's College Hospital in London (UK) for 7–10 days, to determine the seizure focus for possible surgical resection[23]. All patients gave their written informed consent to participate in this study, which was approved by King's College Hospital Research Ethics Committee. Each electrode probe had a total of nine microwires at its end, eight active recording channels and one (low impedance) reference. The electrodes were implanted bilaterally in the hippocampus (24 probes) and amygdala (12 probes). Electrode locations were based exclusively on clinical criteria and were verified by MRI or CT co-registered to preoperative MRI. Two patients were recorded using a 64-channel Digital Lynx system (Neuralynx), with the differential signal from each channel filtered between 0.1 and 9000 Hz, and sampled at 32,556 Hz. The other four patients were recorded using a 64-channel Neuroport system (Blackrock Microsystems), with the differential signal from each channel filtered between 0.3 and 7500 Hz, and sampled at 30,000 Hz.

As in previous works[23,62], a simple visual task was used to identify responsive stimuli. The subject was sitting facing a laptop computer where a set of about 100 stimuli were presented, six times each in pseudorandom order using a block design (i.e., if $N$ stimuli are used in the session, all stimuli will be shown once, in random order, after the first block of $N$ trials, twice after the first $2*N$ trials, etc.). Each trial started with a fixation cross on the screen for 500 ms, followed by a picture displayed for 1000 ms. Then, the screen went black and the patient had to press a key to respond whether or not there was a person in the picture. The inter-trial interval varied randomly between 600 and 800 ms. These "screening sessions" typically lasted about half an hour. The set of pictures used includes familiar items to the patient, such as images of celebrities, landmarks, animals, and the patient's relatives and friends.

Once it had been identified which picture/s triggered the firing of which neuron, we performed follow-up sessions, in which a subset of about 15 stimuli (mean ± std, 13.9 ± 4.5) from the screening session (including all those that elicited a response) were used, but each of these images was shown 25–35 times in pseudorandom order. The data reported here come from these follow-up sessions.

**Single neuron responsiveness criteria**. The collected data were processed offline, and the high-frequency activity (above 300 Hz) was extracted to identify the spikes of the recorded neurons. Spike detection and sorting was done with Wave_clus[63]. In order to assess whether a particular unit was responsive to a certain picture, the following response criterion was implemented: (i) the instantaneous firing rate had to cross over a threshold for at least 75 ms (with the upwards crossing defined as $t_{THR}$, with short periods of less than 20 ms going below threshold being disregarded. The instantaneous firing rate was calculated by convolving the spike train with a Gaussian kernel with $\sigma = 10$ ms (truncated at 1% amplitude). The threshold was set to the mean plus 4 standard deviations, computed across all stimuli between 900 and 100 ms before stimulus onset (with a minimum at 5 Hz, for neurons with low baseline firing); (ii) the median number of spikes (across trials) in a 500 ms window from $t_{THR}$ was at least 2, and larger than the mean plus 5 standard deviations (across all stimuli) of the baseline activity, defined as the median number of spikes (across trials) between 200 and 700 ms before stimulus onset; (iii) the $p$-value of a one-sided paired sign test between the spike count on each trial for the post stimulus (500 ms window from $t_{THR}$) and baseline (200 and 700 ms before stimulus onset, for the particular picture) was less than 0.01.

From the 609 units recorded in the follow-up sessions, the response criterion led to a set of 165 responses in 81 units (11 from the amygdala, 70 from the hippocampus; see Supplementary Table 1 for the distribution of units per experimental session) that matched closely what was visually identified as significant responses. Based on a previously used criterion[64], we classified single units based on: (i) the spike shape and the variance of the cluster; (ii) the ratio

between the peak value of the mean waveform and the standard deviation at their first sample was larger than 5; (iii) the ISI distribution of each cluster; and (iv) the presence of a refractory period for the single units, i.e., <1% spikes with an ISI smaller than 3 ms. This way, we identified 8 out of the 81 units as multiunits, with the remaining 73 classified as single units. In addition, $t_{THR}$ was defined as the spike response onset. We defined as multi-responsive a unit showing at least two significant responses. There was a total of 37 multi-responsive units (7 from the amygdala, 30 from the hippocampus), which responded to an average of 3.3 pictures (s.d: 1.94). 3 out of the 37 units where classified as multiunits (but they did not show significant difference in response strength following the decoding analysis). By identifying the responsive pairs of stimuli in each of these units, we defined a total of 208 responsive pairs. Only 5 of these pairs were coming from multiunits.

**Quantification of spike activity**. The strength of spike activity (in Hz) was defined as the median number of spikes (across trials) fired by a unit in a given time window, normalized by the window length (Figs. 1, 2). During the baseline period, it was computed between 1000 and 300 ms before stimulus onset; whereas the strength of individual responses was calculated in a 700 ms window starting at the spike response onset.

For each multi-responsive neuron, the strength of response was obtained by pulling together all the trials from the different stimuli eliciting a significant response and calculating the median number of spikes across them. In this case, the response window started at the minimum spike response onset across responses.

To quantify the normalized strength in Fig. 2c, first we measured, for each responsive unit, the median number of spikes across trials between 100 and 800 ms after stimulus onset for every stimulus. Then we corrected the activity by the mean baseline (across all stimuli) and normalized by the maximum across all stimuli. When a unit showed activity below baseline for a certain stimulus, it was assigned a zero strength.

**Quantification of differences in response strength**. For each multi-responsive neuron, a surrogate test was performed for pairwise comparison of the strength of spike response (Figs. 1, 2). Given a pair of significant responses from the same unit, we compared their absolute difference in strength of response $\Delta S_r$ to a distribution of 1000 surrogate values, created by randomly permuting the trial labels for the two responses. Specifically, for each re-arrangement of the labels, we obtained two surrogate responses and calculated their absolute difference in strength of response ($\Delta S_i$). The ranking of the real value ($\Delta S_r$) among the population of surrogate values, gave the p-value for the null hypothesis that the two responses had the same strength.

An analogous surrogate test was implemented to compare the average spike shapes of the spikes fired in response to the different pictures eliciting responses in the same neurons (shuffling the response labels). The shapes of spikes associated to different responses were compared using as test statistic the Mahalanobis distance between the two groups of spikes fired in the response window between 100 and 800 ms after stimulus onset. This test was implemented in order to assure that the multiple responses come from the same neurons and are not due to spurious spike sorting.

To assure that the relatively small number of trials used is enough to obtain reliable estimates, we also assessed how the differences in response strength for multiple responses in the same neurons changed with the number of trials used in the estimation (Supplementary Fig. 2). For a given number of trials ($n_t$), the strength of the single response was defined as mean number of spikes fired in a fixed response window (100 to 800 ms) across trials 1 to $n_t$. For each pair of responses (208 pairs in total), we estimated the absolute strength difference as a function of $n_t$, normalized by the maximum strength in the pair as described below in "Association scores between stimuli". Finally, we computed the grand average across all pairs and fitted the data points with a negative exponential function to verify the existence of a plateau (i.e., if the estimation remained stable after a certain number of trials $n_t$).

**Decoding analysis**. A naive Bayesian decoder with leave-one-out cross-validation was run on each multi-responsive unit to test whether the identity of the responsive stimuli could be predicted based on the single trial spike count in the response period (Figs. 1, 2). The decoding performance was estimated as percentage of trials correctly predicted, and its statistical significance was assessed in comparison to the performances obtained on a population of 1000 surrogates created by randomly shuffling the trial labels.

**Study of single neuron response latencies**. We used a surrogate test implementation similar to the one used for response strength to compare the response latencies of each pair of significant responses from the same unit. For this, we calculated the difference in latency of response ($\Delta L_r$) for each response pair and, as above, for each comparison the p-value was obtained by comparing the real test statistic value with 1000 surrogate values.

We also examined whether the estimation of the spike response latency changed with the number of trials (Supplementary Fig. 4). First, for each of the 165 responses we estimated the response onset (as described above in "Single neuron

responsiveness criteria") considering a number of trials $n_t$, i.e., from trials 1 to $n_t$. The average onsets across all the responses as a function of $n_t$ were fitted with a negative exponential function.

Finally, we estimated the response onsets on the whole population considering a sliding window with a fixed number of trials ($n_t = 6$) moving from the beginning to the end of the trial list (i.e., if $N$ is the maximum number of trials for the given response the window starts in trials 1 to 6 and slides to trials $N$-5 to $N$). A two-sided paired sign test was used to compare the estimation on trials 1 to 6 with the one on trials 20 to 25 to show that the estimation bias was related to the length of the window and not to the number of repetitions the stimuli had been presented.

**Local field potentials**. The raw data were filtered between 2 and 512 Hz (zero-phase elliptic filter). A notch filter (2nd order IIR) was used to remove 50 Hz line noise and its harmonics. Finally, the signal was downsampled to 1.5 KHz. We computed the power spectrum for each of the recorded channels and discarded those exhibiting very large high frequency noise (5 out of the 69 channels with at least one spike response). As a result, 151 spike responses, 33 multi-responsive units, and 192 responsive pairs, had an associated usable LFP. The single-trial LFP traces were extracted from 1 s before to 2 s after stimulus onset. For the analysis in the theta range, the LFP was further filtered between 3 and 6 Hz (zero-phase elliptic filter). Each trial was normalized by the squared root of the mean squared signal between 0 and 1000 ms after stimulus onset.

Instantaneous power of the evoked LFP was computed using the squared magnitude of the Hilbert transform of the average LFP in the theta range (Fig. 3). Each response (LFP power or firing rate) was normalized by the area below the curve before computing the grand averages. LFP latency was defined as the time when the instantaneous power of the evoked LFP crossed a threshold, set as the median plus 4 mean absolute deviations, computed across all stimuli between 900 and 100 ms before stimulus onset.

To compare multiple LFP responses, we used a similar surrogate test implementation (shuffling labels), as described above. Specifically, we compared the peak latency of the LFP responses, which we estimated by searching for the maximum of the instantaneous theta evoked power between 0 and 700 ms after stimulus onset.

**Phase locking analysis**. For each response, we considered the spikes in two time-windows of length equal to half a cycle of the mid frequency in the theta range (i.e., ~111 ms): "baseline" (time window in the baseline period) and "response" (time window starting at the spike response latency) (Fig. 5). Given than in several cases there were too few spikes, in the "baseline" epoch we considered the spikes from all the stimuli. We asked for a minimum of 20 spikes in the baseline/response window to include a response in the analysis. At each spike time, we computed the instantaneous phase using the angle of the Hilbert transform of the single trial LFP filtered in the theta band (with 0° and ± 180° representing the peak and trough of the oscillation, respectively). For each spike response and epoch, we computed the mean resultant vector (with its magnitude being the phase locking index, PLI, and its angle the circular mean of all the angles in the epoch) and performed a Rayleigh test (evaluating the hypothesis of a uniform phase distribution).

For the surrogate analysis, spike time was left unchanged whereas the single trial LFPs were shuffled, and the PLI was recomputed based on the shuffled phases. This way, the same number of phases were used on each surrogate making it fair to compare the resulting PLI values.

For the analysis of multiresponses, we considered the 44 cases where both responses in the pair were significantly phase locked to the LFP filtered in the theta band (these cases were based on 60 different individual mean vectors). For each of these pairs we calculated the angular difference ($\Delta\theta_i$) between the two mean phases associated to each of response.

**Web association scores between stimuli**. As in our previous work[22], we used a web-based metric to measure the strength of association between stimuli (Fig. 6). Using an internet search engine (Bing), for each pair of stimuli, the strength of association between them was evaluated as:

$$a_{ij} = \log_2\left(\frac{\text{hits}(\text{concept}_i \text{ AND } \text{concept}_j)}{\text{hits}(\text{concept}_i) \cdot \text{hits}(\text{concept}_j)}\right), \tag{1}$$

where hits($\cdot$) represents the number of hits (pages containing the searched concept) given as result of the web search, while the use of the AND operator gives the number of pages containing both searched concepts. Note that as the measure is normalized, it is not affected by popularity or current relevance. A script was used to avoid the search history to affect the results. The web search was limited to famous people and places, excluding names of family members and animals. The values for each recording session were normalized using a z-score. This web-based association score was shown to successfully replicate the results using personal scores directly assigned by the patients[22].

For each multi-responsive neuron, we defined an average association score between all the pair of images eliciting responses ($AS_{R-R}$), as mean of $a_{ij}$ across all possible pairs of responsive stimuli in the session. Similarly, for each unit the association score between responses and non-responses ($AS_{R-NR}$) was obtained as

mean of $a_{ij}$ on all the pair of images where one elicited a response and the other did not. The scores $AS_{R-R}$ and $AS_{R-NR}$ were compared across units using a one-sided paired sign test.

Considering all the pairs of significant responses with a valid association score $a_{ij}$ (139 pairs from 35 units), we calculated the Pearson correlation coefficient ($r$) between the normalized differences in strength of response and the corresponding association scores (Fig. 6). For each pair, the normalized difference in strength was obtained as the absolute value of their strength difference divided by the highest strength value in the pair. Each response strength was computed as the mean of the spike count instead of the median, as the latter led to a figure with more discrete values. Still, the correlation using the median produced similar results (Pearson correlation, $r = -0.17$, $p = 0.04$).

The same analysis was used to assess the correlation between absolute differences in spike latencies and association scores on responsive pairs, and between absolute difference of LFP latencies and association scores on responsive pairs.

**Personal association scores between stimuli.** In 10 out of 21 experimental sessions, subjects were asked to fill a "personal association matrix", in which they ranked between 0 and 10 how much a subset of approximately 10 pictures were related to each other. Entries given by the subjects were normalized with a z-score. The subset of the stimuli comprised images eliciting responses in the recorded neurons, as well as other non-responsive pictures presented in the experimental session.

**Semantic categories, familiarity and visual similarity.** We defined 8 different categories to associate with the individual stimuli: musicians, experimenters, actors, sportsmen, politicians, places, objects, and animals. For each unit, we looked for all the category pairs and compute the associated scores $AS_{R-R}$ and $AS_{R-NR}$. For example, if we had a response to two actors, we compared the association score between them to the ones between other actors; if we have another neuron firing to an actor and a place, we compared the association score to the ones of other actors and places.

To evaluate stimulus familiarity, we calculated the product of the number of hits obtained for each independent search and defined

$$f_{ij} = \log_2 \left( \text{hits}(\text{concept}_i) \cdot \text{hits}(\text{concept}_j) \right) \tag{2}$$

and then computed the scores for the R–R and R–NR pairs. Likewise, we controlled for perceptual similarities between pictures by estimating the visual similarity $v_{ij}$ for each pair of stimuli as the cross-correlation between the images (each with $160 \times 160$ pixels, grayscale and z-score-normalized).

**Strength and latency in relation to the number of responses.** The whole set of 81 responsive units was grouped according to the number of stimuli eliciting a response, i.e., number of responsive stimuli (Fig. 7). The correlation between the spike response latencies and the number of responsive stimuli on each unit was assessed with a Spearman's rank correlation test. In the same way, we tested the correlation between the response strength and the number of responses on each unit. As in Fig. 6b, the strength was computed using the mean instead of the median, although the correlation using the median produced similar results (Spearman correlation, $\rho = 0.39$, $p \sim 10^{-6}$).

**Code availability.** The main codes used to generate the results of this work can be downloaded from https://www2.le.ac.uk/centres/csn/software.

## Data availability
The data that support the findings of this study are available from the corresponding author upon reasonable request.

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

## Acknowledgements

The authors thank all patients for their participation and the King's College Hospital staff for technical assistance. This work was supported by grants from the Medical Research Council (G1002100) and the Human Frontiers Science Project. We want to thank Edmund Rolls for the valuable feedback and discussions.

## Author contributions

Experimental design and project supervision, H.G.R. and R.Q.Q.; formal data analysis, H.G.R., E.D.F.; contribution to data analysis, M.J.I.; software, data curation H.G.R., E.D.F.; data collection, H.G.R., E.D.F, M.J.I.; resources, A.V., G.A., M.P.R., R.Q.Q.; funding acquisition, H.G.R., R.Q.Q., M.P.R.; R.S. performed the surgeries; writing—original draft, H.G.R., E.D.F. and R.Q.Q.; writing—review and editing, all authors.

## Additional information

**Competing interests:** The authors declare no competing interests.

