## [Peer Review File · Nature Communications]

Reviewers' comments:

Reviewer #1 (Remarks to the Author):

In this manuscript, Dr. Quiroga and colleagues examine a long-standing and important question in neuroscience — how are memories encoded at the level of individual neurons in the human temporal lobe. To study this, they use an inventive two-step process. In the first, they present human participants with a large array of images as they record from neurons in their MTL. Second, they repeat the task but with many more repetitions and with images that neurons are responsive and which may be associated to these primary images. Overall, they find that even though the associated images may be quite different in presentation from these primary images, neuronal responses remain quite similar. They use a large number of analyses and appropriate controls to demonstrate this and to rule out a number of possible confounds. Finally, they perform a spike to field comparison that reveals a latency of response that reflects the degree of association between images. Based on these observations, they conclude that these neurons follow a 'neural unitization' principle whereby cells respond similarly to stimuli that are closely associated and argues against a 'grandmother cell' type representation whereby each cell represents a unique stimulus or concept.

Overall, the authors did a commendable job at studying an important and timely question in neuroscience. They also use a fairly unique approach in which they first search for responsive cells and then test them with a more 'focused' array of associated images. While I like this paper and would hope to see it ultimately published, there are a number of confounding issues especially with regards to its interpretation and conclusions.

First, association between images needs to be better quantified. For example, the association between the fact that "Leslie Nielsen had acted in a 1980 film that involved an airplane and a picture of an airplane cabin" is fairly loose and not well quantified. My guess is that most people would not innately make such a connection. While the authors use a web search engine-based approach to quantify similarity, such an approach is somewhat unusual from the neuroscience perspective. For example, quantified associates can be biased by recency of search, current trends in the news, etc. The results are also contemporaneous to the time and location of search. One potential way to address this, is to ask the participants to rank the similarity between images as well as rank their familiarity with them (see also below). Such results can be potentially obtained by the research team retrospectively and would not require additional recordings.

It is not clear to me how this paper is referencing a memory process rather than simply categorization. In my view, the limited variability in neuronal response observed here is consistent with a categorization signal. Overall, the current task does not require the participants to learn or recall associated images. Rather, it is based on a largely passive recognition/identification of images. A large number of studies in primates (DiCarlo) and humans (Fried) have already demonstrated categorization responses, which also display similar responses to different images that are associated by category. One potential way to address this is to perhaps examine responses to images (within your current data set) that can be more easily categorized based on standard classifiers (i.e., inanimate vs. animate objects, etc).

While I liked the authors two-step approach, it is possible that selection of units and then images based on their responses sets up a 'selection bias'. In other words, it is possible that by selecting neurons that were highly responsive to particular test images, the authors were ignoring cells with intermediate responses or neurons that are more likely to display mixed-selectivity. While the authors did test for differences in strength of response across cells that had already been tested, this still does not address this selection bias concern. I don't know that I have a good suggestion of how to address this.

While there has been some debate within the field on how well the recording approach used by the authors can truly identify well-isolated cells, it is important to further quantify whether and to what extent multi-units were present. While the authors provide some analysis and indicate that they are confident that they selected only single-units, having multi-units in their data would significantly confound their results and conclusions. In other words, multi-units would almost guarantee that they will not find highly response-selective cells.

Overall, the paper is quite promising and hopefully these suggestions can help.

Reviewer #2 (Remarks to the Author):

The study by Rey and colleagues aimed to better understand the manner in which human hippocampal neurons represent declarative memories, and associative memories in particular. To address this question, the authors recorded from single neurons in patients undergoing surgery for epilepsy, and then characterized the response profiles of neurons that responded to more than one stimulus. Results indicated that a given neuron's response (strength, latency, etc.) was highly similar to each of these stimuli, leading the authors to conclude that hippocampal neurons represent each stimulus comprising an associative memory in a unitized manner.

This study involves data that is challenging to acquire and reflects an impressive amount of analytical work on the authors' part. However, as described in greater detail below, I have concerns regarding the framing of the paper, conclusions drawn from the data, and apparent circularity of a subset of analyses used.

1. I find the framing of the study – whether hippocampal neurons represent stimuli in a graded fashion, similar to tuning curves in cortical sensory areas – to be inappropriate given the methods used. A more appropriate framing would seem to be: How do hippocampal neurons represent associated stimuli? This is a perfectly interesting question that builds upon recent findings from the same group (de Falco et al., 2016), and allows them to characterize in depth the “unitized” manner in which a neuron responds to items comprising previously learned associations. This framing is quite different from investigating whether hippocampal neurons exhibit tuning curves such that they are most responsive to a target stimulus and progressively less responsive as stimuli become less similar to the target. Answering this question would seem to require experimental manipulation of stimulus similarity along some dimension – visual, semantic, etc. I fully agree with the authors' statement (line 413) that it would be challenging to pick a feature dimension that would allow for this type of testing, but nonetheless if they frame their paper as testing for the existence of tuning curves, then certainly there should be an attempt to look at responses to stimuli varying along some dimension. A recent study by Suthana et al (2015), for example, identified images to which neurons responded, and then in a subsequent recording session, presented images of varying similarity to the images that initially elicited a response. The focus of that paper was not to assess the hippocampal tuning curve idea, but nonetheless the methodology used is more similar to what I expected to see in the current study upon reading the framing and aims. The only aspect of the current study that seems to be at least somewhat well-suited to addressing the graded response idea is the use of an internet-based association rating between pairs of stimuli. By looking at neuronal responses to pairs of stimuli that range in association strength, the authors demonstrated that there is indeed a graded pattern, such that responses are very similar when the association between two stimuli is strong, and less so when the association is weaker. This finding, however, seems to be at odds with how they frame their results – the topic of my second point.

2. The authors interpret their findings as refuting the idea that hippocampal neurons exhibit graded responses similar to a tuning curve, and instead conclude that, when neurons respond to more than one stimulus, they respond equally strongly in a “unitized” manner. However, they also highlight that the more closely related two stimuli are, the more similar the neuronal responses to

these images – which seems to imply a graded response. I realize that the latter results pertain to responses across many different units rather than within the same neuron, but nonetheless, it is challenging to reconcile these two ideas. These mixed messages are readily apparent in the abstract: “In contrast to the graded responses ubiquitously observed in cortex, we found that most of these neurons exhibited no differences in their spike and local field potential responses to the individual stimuli. Moreover, the similarity of the neural responses correlated with the degree of association between stimuli.” How do the authors relate the latter finding to their conclusion that neuronal responses are not graded along some feature dimension (e.g., semantic relatedness)?

3. Finally, I found the analyses pertaining to Figures 1 and 2 to be a bit circular. Specifically, “multi-responsive” neurons were defined as those that met criteria for responding strongly to more than one stimulus. If a given neuron is defined as “multi-responsive” because it responds strongly to both A and B, it seems circular to then perform statistical analyses showing that the response to these two stimuli was very similar. It seems important, when creating the criteria for determining whether neurons are responsive, to use liberal thresholds that would then allow one to later test for differences in responsiveness. If, instead, a conservative set of criteria is used, then by definition the response to both stimuli has to be very strong; not surprisingly, when one then compares these responses, they are similar. Figure 2b helps to address this issue somewhat by showing the distribution of response strength for responsive units, but this analysis seems circular as well, in that it is comparing distributions for stimuli that did and did not elicit responses. Again, not surprisingly, when these distributions are compared, the distribution for “responses” has mostly high strength responses, and the distribution for “non-responses” has mostly low strength responses. It would be more helpful to see the see this distribution without the separation between responses and non-responses.

Minor concerns:

1. The term “unitization” has clear meaning in the memory literature – the process of creating an integrated representation that includes all elements/items of an association. If space allows in the Introduction, it would be useful to include a definition and a few references to prior studies of unitization to familiarize readers with this concept.

2. I found the term “responsive pairs” of stimuli to be confusing. The stimuli themselves weren't responsive, but rather, these stimuli elicited responses from neurons. When reading the manuscript, it was at times challenging to understand what “responsive pairs” referred to (i.e., pairs of stimuli, pairs of responses, etc.). To reduce this confusion, I suggest instead using the term “response-eliciting pairs” of stimuli.

Reviewer #3 (Remarks to the Author):

This manuscript reports single unit data from a modest number of neurons in human amygdala and hippocampus. The main findings are that some units were responsive to multiple stimuli, and that key features of the responses to the driving stimuli were similar and showed modest (but significant) correlations with an association strength metric. From these data, the authors speculate that the MTL neural code for these pairs/sets of stimuli is ‘unitized’.

While the questions addressed are of potential interest, there are a number of dimensions of the manuscript that raise concerns. The main concerns are detailed below, with the hope that they will be helpful to the authors.

1) The manuscript's framing is ineffective. First, the Introduction frames the question around the broad issue of what is the nature of the neural code in the MTL. This is such a vast question, and unfortunately the Introduction does not situate the work within the rich literature on this topic

(e.g., complementary learning systems theories; the wealth of data on how stimulus and environmental stimuli that vary is similar given rise to continuous vs. sigmoidal responses that bear on claims about pattern separation vs. pattern completion; the animal and human literature on memory integration, unitization, and integrative encoding; etc). Moreover, the notion of an 'association' is introduced in the Abstract and Introduction, but it is not until well into the results that the reader comes to realize that the work relates to the sub-literature on the relationship between pre-existing knowledge/concepts and MTL coding.

I would recommend that the authors completely re-frame the Abstract and Introduction, narrowing it rather to the set of questions that has emerged about the role of pre-existing associative knowledge and MTL coding. I would then conclude the Introduction with a set of hypotheses to be tested that are specific to this aspect of the experiment.

2) Stimuli during the follow up session were presented in a 'pseudorandom order'. What is meant by 'pseudorandom' (vs. random) is unclear. More critically, given evidence of rapid sequential/statistical learning effects within the hippocampus, it seems important to compute the first and second order transition probabilities (and perhaps even longer trial history effects) between stimuli and to then examine whether the probability of a 'unitization' effect also relates in any way to the transition probabilities. Note that by including transition probabilities as covariates, this may also increase sensitivity to pre-experimental associative effects.

3) As the authors state, the association score that was computed is a measure of pre-experimental semantic relatedness, rather than a measure of episodic association.

- Note that this effect appears rather weak. Why use a one-sided paired sign test? The fact that this is a one-sided test should be stated in the results text; currently this only appears in the figure caption.
- Given that this score measures pre-experimental relatedness, what accounts for why the multi-responsive neurons did not originally respond to both members of a pair of stimuli that ultimately were classified as R-R during the follow up period of the experiment. Were all stimuli not included during stimulus selection? If not, how were additional stimuli selected for inclusion in the follow up phase of the experiment? If so and if these pairs were already linked in semantic memory, why was repeated repetition required to elicit a similar response in the small population of amygdala and hippocampal neurons identified as multi-responsive here?
- It would be helpful to examine the temporal profile over which the multi-responsive units came to demonstrate such responses. That is, was the effect evident immediately at the start of follow up or did it take time to emerge. If immediately at the outset of follow up, did this differ in any way from the stimulus selection phase of the experiment?
- For the correlations plotted in Figs 6b-d, please also plot the corresponding correlations for the R-NR pairs, and please test and report whether the strength of the R-R correlations significantly differed from that of the R-NR correlations? Moreover, related to the point about the effect size, given the small amount of variance explained by the association score, the following conclusion seems to overstate the data: "the degree of association appears to be the key metric underlying the neuronal coding in the MTL". No other behavioral or stimulus dimensions were explored during data analysis, and so such a conclusion does not seem justified. Note also, given that the hippocampus can rapidly form relational/associative/conjunctive representations of previously unassociated pairs stimuli (as is stated in the Discussion), such a statement (which appears to apply to pre-existing associations) appears inconsistent with a vast literature.

4) The notion of 'unitization' has pre-existing meaning in the literature on MTL coding (e.g., see work from Ranganath and colleagues). In that past work, the 'unitization' hypothesis led to articulation of clear behavioral criteria had to be met for two stimuli to be viewed as 'unitized'. In the present work, not such criteria are specified. As such, it is unclear what the authors mean by 'unitization' here, nor whether there is independent behavioral evidence supporting such a claim (beyond the similar neural responses reported herein).

Minor comments

- a) From the methods, it is unclear what percentage of the units in the hippocampus and in the amygdala failed to meet the criteria for being 'responsive to a certain picture'.
- b) Please report the number of units in hippocampus and amygdala that were responsive in each subject. Moreover, while the number of multi-responsive units in each region was small, it seems important to at least provide a qualitative statement as to whether there were regional differences of any kind in the degree of response similarity as a function of association strength. If no differences are evident, do the authors argue that the amygdala codes for concepts in the same way that they are arguing for concept coding in hippocampus?
- c) The methods and results reporting are rather confusing; this relates, in part, to the challenging framing of the manuscript (as I read the results and methods, I found myself repeatedly trying to figure out why particular analyses were being performed and over which aspects of the data). As one critical example, when stating that 37 of the 81 'responsive units' were 'multi-responsive', was this during the initial recording/stimulus selection period or during the follow up repeated presentation period? I assume the latter, but this is difficult to discern early on when reading the methods and results.

RESPONSE TO THE REVIEWERS' COMMENTS

We thank the reviewers for the very useful feedback, which we believe have helped us to improve our manuscript.

Reviewer #1 (Remarks to the Author):

In this manuscript, Dr. Quiroga and colleagues examine a long-standing and important question in neuroscience – how are memories encoded at the level of individual neurons in the human temporal lobe. To study this, they use an inventive two-step process. In the first, they present human participants with a large array of images as they record from neurons in their MTL. Second, they repeat the task but with many more repetitions and with images that neurons are responsive and which may be associated to these primary images. Overall, they find that even though the associated images may be quite different in presentation from these primary images, neuronal responses remain quite similar. They use a large number of analyses and appropriate controls to demonstrate this and to rule out a number of possible confounds. Finally, they perform a spike to field comparison that reveals a latency of response that reflects the degree of association between images. Based on these observations, they conclude that these neurons follow a 'neural unitization' principle whereby cells respond similarly to stimuli that are closely associated and argues against a 'grandmother cell' type representation whereby each cell represents a unique stimulus or concept.

Overall, the authors did a commendable job at studying an important and timely question in neuroscience. They also use a fairly unique approach in which they first search for responsive cells and then test them with a more 'focused' array of associated images. While I like this paper and would hope to see it ultimately published, there are a number of confounding issues especially with regards to its interpretation and conclusions.

We appreciate the comments from the reviewer on the general assessment of our manuscript and the methodology we have used, and we answered each of the detailed comments below.

1) First, association between images needs to be better quantified. For example, the association between the fact that "Leslie Nielsen had acted in a 1980 film that involved an airplane and a picture of an airplane cabin" is fairly loose and not well quantified. My guess is that most people would not innately make such a connection. While the authors use a web search engine-based approach to quantify similarity, such an approach is somewhat unusual from the neuroscience perspective. For example, quantified associates can be biased by recency of search, current trends in the news, etc. The results are

also contemporaneous to the time and location of search. One potential way to address this, is to ask the participants to rank the similarity between images as well as rank their familiarity with them (see also below). Such results can be potentially obtained by the research team retrospectively and would not require additional recordings.

In the patient cohort presented in this study, we were able to obtain the personal association scores in about half the sessions, where subjects ranked how different pairs of stimuli were associated (from 0 to 10). The following figure shows the result of correlating the web-based and personal association matrices after a z-score normalization, which we include as Supplementary Fig. 6a in the new version of the manuscript.

As it can be seen in the figure, there is a strong correlation for highly associated items and more variability on the web score for those that are less associated. This is due to the fact that there is more general agreement about things that are highly related (which is also reflected in the web-search metric), whereas results for less associated concepts are more prone to subject-to-subject variability given by the subjects' personal experiences. As the reviewer noted, some subjects may know and some others may not know the 80's movie linking Leslie Nielsen to an airplane. Overall, there was a significant correlation between the metrics (Spearman correlation, $\rho = 0.24$, $p = 5.9 \times 10^{-3}$). This is in agreement with the result presented in De Falco et al (2016) using a larger patient cohort and supports the use of the web-association scores as a proxy for the personal scores from the patients.

In addition, the search was done using a dedicated API to create scripts that can avoid biasing the results based on search history. It should also be noticed that all the pictures shown in the experimental session were related to concepts that were highly familiar to the subject.

2) It is not clear to me how this paper is referencing a memory process rather than simply categorization. In my view, the limited variability in neuronal response observed here is consistent with a categorization signal. Overall, the current task does not require the participants to learn or recall associated images. Rather, it is based on a largely passive recognition/identification of images. A large number of studies in primates (DiCarlo) and humans (Fried) have already demonstrated categorization responses, which also display similar responses to different images that are associated by category. One potential way to address this is to perhaps examine responses to images (within your current data set) that can be more easily categorized based on standard classifiers (i.e., inanimate vs. animate objects, etc).

The reviewer correctly points out that there is no active memory task in our experiment. We believe this is a very interesting point, because we interpret our results as correlates of long-term memory coding in the hippocampal formation. As we now discuss in the new version of the manuscript (in the Introduction and Discussion sections) the current findings markedly contrast with a previous finding in which we indeed have an active memory task (subjects learned paired associates) and which did not show any unitization. Putting together these pieces of evidence, we argue that unitization is only present after learned associations are consolidated.

We believe our data cannot be explained as a categorization process and have done further analysis to support this claim. We should first note that when neurons show responses to particular categories, they do so by responding significantly stronger to the items within the category than to those in other categories. We have mentioned some of the works from the Di Carlo group supporting this idea (Hung et al. 2005; Kreiman et al 2006). However, if a neuron responds preferentially to stimuli belonging to a specific category, it does not necessarily respond to these stimuli with the same strength and latency. In fact, this is not the case for category responses in the monkey inferotemporal cortex. In particular, Figure 1A in Hung et al. 2005 shows examples of multi-responsive units, exhibiting clear differences between the responsive stimuli in both strength and latency, and Kreiman et al 2006 showed that image selectivity within a category was observed in both spike and LFP responses (Figure 3E). In contrast to these studies, we here show a nonsignificant decoding performance for most of the neurons with multiple associated responsive stimuli (i.e. in most neurons responses were indistinguishable from each other). We have now clarified this in the Discussion.

To provide further evidence that our results cannot be interpreted as category responses, following the reviewer's comment we defined 8 semantic categories to which our stimulus set belonged: experimenters, actors, musicians, sportsmen, politicians, places, objects, and animals. For each unit, we looked at the association scores for all the response pairs AS_{R-R} (see Online Methods) and compared them to the associated scores with other pictures the neuron did not respond to AS_{R-NR} , but constraining the comparison to pictures within the same categories. For example, if we had a response to two actors, we compared the association score between them to the ones between other actors; if we have another neuron firing to an actor and a place, we compared the association score to the ones of other actors and places. This way, if neurons are just showing category responses, we expect no difference between AS_{R-R} and AS_{R-NR} , whereas if we do find a difference, this means that neurons fire to specific associated pairs and not to broad category of stimuli. The resulting values can be seen in the following figure (also in Supplementary Fig. 6b).

We found a significant difference between AS_{R-R} and AS_{R-NR} ($n=85$, paired sign test, $p < 0.05$), thus supporting the view that results cannot be just seen as broad semantic category responses.

To rule out other potential confounds, we also showed that it was not related to stimulus familiarity, where we calculated the product of the number of hits obtained for each independent search and defined (see Online Methods) the score

$$f_{ij} = \log_2(\text{hits}(\text{concept}_i) \cdot \text{hits}(\text{concept}_j))$$

In fact, we found no significant difference between this score for R-R and R-NR (paired sign test, $p = 0.63$). The scores can be seen in the following figure (also in Supplementary Fig. 6c).

In addition, we tested the possibility that perceptual similarities between pictures could explain the appearance of response-eliciting pairs. To rule out this confound, we estimated the visual similarity v_{ij} for each pair of stimuli as the cross-correlation between the images (each with 160 x 160 pixels, greyscale and z-score-normalized). Again, we found no significant difference between this score for R-R and R-NR (paired sign test, $p = 0.74$). The scores can be seen in the following figure (also in Supplementary Fig. 6d)

All these results are in line with the ones obtained in De Falco et al. 2016 using a larger dataset.

3) While I liked the authors two-step approach, it is possible that selection of units and then images based on their responses sets up a 'selection bias'. In other words, it is possible that by selecting neurons that were highly responsive to particular test images, the authors were ignoring cells with intermediate responses or neurons that are more likely to display mixed-selectively. While the authors

did test for differences in strength of response across cells that had already been tested, this still does not address this selection bias concern. I don't know that I have a good suggestion of how to address this.

The reviewer raises an important issue, namely a potential bias due to the selection of pictures included in the experiment and the selection of units included in the analysis. We analyse these two cases in turn.

With respect to the picture selection bias, we used the initial screening session, in which about 100 pictures were shown 6 times each, to choose a subset of about 15 pictures (including those eliciting responses) to use in the follow up experimental session in which we report unitized responses (we cannot do it with the first set because 6 trials are not enough to statistically compare responses). Now, the reviewer is correct that if in such selection process we would have discarded pictures eliciting intermediate responses, we would have introduced a bias towards finding similar responses in the follow up experiment. To show that this was not the case – i.e. that we did not discard intermediate responses in the first set – we did the following analysis.

In the initial screening session, we had a set of pictures eliciting responses that were included in the follow up session. The remaining pictures (to get up to 20) were selected from the ones not eliciting responses in the initial screening. Let us call FU, the set of pictures (not eliciting responses) that were selected to be used in the follow-up session and NFU the ones that were not selected to be part of the follow-up session. For each of the 980 recorded units in the initial screening sessions we sorted the spike count for the FU and NFU sets independently. Then, if there were N_i nonresponsive stimuli in the FU set, we chose the N_i stimuli with the largest spike count in the NFU set, and compared the average spike count (subtracting the average baseline spike count) for both sets. The results can be seen in the following figure.

Comparing the spike count in the FU and NFU sets led to no significant differences (ranksum test, $p = 0.33$), with medians equal to 0. These results support the fact that nonresponsive stimuli had a spike count no different than baseline activity, which was also no different between the FU and NFU sets, thus showing that we did not miss intermediate responses in the picture selection procedure.

Second, we explored the possibility of bias given by excluding neurons with intermediate responses. To address this issue, we focused on the data from the follow-up experimental sessions and explored the results using different responsive criteria (i.e. including neurons with weaker responses).

Particularly, in our response criterion we ask for: *“the median number of spikes (across trials) in a 500 ms window from t_{THR} was larger than the mean plus 5 standard deviations (across all stimuli) of the baseline activity...”*. We defined a parameter K for the number of standard deviations that the response need to deviate from the mean baseline firing, and defined response sets for $K = 2.5, 4, 5$, and 6 ($K = 5$ is the set presented in the manuscript). For each set, we computed the strength and spike latency differences for all the “response-eliciting pairs” as in Figures 2A (now 2B in the revised version of the manuscript) and 4A.

First, we checked if results were different when considering the more restrictive criterion of $K = 6$. Specifically, we compared the set of pairs with $K = 6$ ($n = 137$ pairs) with those pairs added when considering $K = 5$ (71 pairs), i.e. the latter being the difference between the 208 pairs from the set with $K = 5$ and the 137 pairs from the set with $K = 6$. We found no significant difference in either strength or spike latency difference (ranksum test, $p = 0.28$ and $p = 0.3$, respectively). Next, we considered the set of $K = 5$ and compared it to the pairs added with $K = 4$ ($n = 68$). Again, we found no significant difference in either strength or spike latency difference (ranksum test, $p = 0.13$ in both cases). This shows that considering slightly stronger or milder responses ($K = 4, 5, 6$) did not change our main finding of unitization. Finally, we took the set of $K = 4$ and compare it to the pairs added when considering $K = 2.5$ (half the value of the original threshold) (127 pairs). In this case, we found the latter set to have significantly larger differences in both strength and spike latency (ranksum test, $p = 4.1 \times 10^{-3}$ and $p = 7 \times 10^{-3}$, respectively), but this is due to the fact that with such relatively low threshold of responsiveness we included many false positives. To illustrate this, the figure below shows some of the “responses” that were included in the set with $K = 2.5$, with their corresponding z-score (values >2.5 are considered a response with this criterion).

Editorial Note: In the figure below, an asterisk indicates an image that has been redacted to protect copyright claims.

4) While there has been some debate within the field on how well the recording approach used by the authors can truly identify well-isolated cells, it is important to further quantify whether and to what extent multi-units were present. While the authors provide some analysis and indicate that they are confident that they selected only single-units, having multi-units in their data would significantly confound their results and conclusions. In other words, multi-units would almost guarantee that they will not find highly response-selective cells.

Our criterion to define single units is now properly explained in the Methods section, which reads:

“Based on a previously used criterion (Quiari Quiroga et al., 2008), we classified single units based on: (i) the spike shape and the variance of the cluster; (ii) the ratio between the peak value of the mean waveform and the standard deviation at their first sample was larger than 5; (iii) the ISI distribution of each cluster; and (iv) the presence of a refractory period for the single units, particularly, less than 1% spikes with an ISI smaller than 3 ms.”

As described in the Methods section, this criterion led to only 8 out of 81 responsive units being classified as multiunit. Moreover, from the 37 multi-responsive units, only 3 were classified as multiunits. Still, the fact that only 5 out of the 208 (2.4%) response-eliciting pairs came from those 3 multiunits shows that the effect of multiunits in our results is negligible.

Overall, the paper is quite promising and hopefully these suggestions can help.

We appreciate the feedback from the reviewer and hope to have adequately clarified his/her concerns.

Reviewer #2 (Remarks to the Author):

The study by Rey and colleagues aimed to better understand the manner in which human hippocampal neurons represent declarative memories, and associative memories in particular. To address this question, the authors recorded from single neurons in patients undergoing surgery for epilepsy, and then characterized the response profiles of neurons that responded to more than one stimulus. Results indicated that a given neuron's response (strength, latency, etc.) was highly similar to each of these stimuli, leading the authors to conclude that hippocampal neurons represent each stimulus comprising an associative memory in a unitized manner.

This study involves data that is challenging to acquire and reflects an impressive amount of analytical work on the authors' part. However, as described in greater detail below, I have concerns regarding the framing of the paper, conclusions drawn from the data, and apparent circularity of a subset of analyses used.

We appreciate the reviewer's feedback and address each of the points below. In general, following the reviewer's comments we have largely changed the framing of the paper, making it more specific and we also performed further analysis to rule out a potential circularity problem.

1. I find the framing of the study - whether hippocampal neurons represent stimuli in a graded fashion, similar to tuning curves in cortical sensory areas - to be inappropriate given the methods used. A more appropriate framing would seem to be: How do hippocampal neurons represent associated stimuli? This is a perfectly interesting question that builds upon recent findings from the same group (de Falco et al., 2016), and allows them to characterize in depth the "unitized" manner in which a neuron responds to items comprising previously learned associations. This framing is quite different from investigating whether hippocampal neurons exhibit tuning curves such that they are most responsive to a target stimulus and progressively less responsive as stimuli become less similar to the target. Answering this question would seem to require experimental manipulation of stimulus similarity along some dimension - visual, semantic, etc. I fully agree with the authors' statement (line 413) that it would be challenging to pick a feature dimension that would allow for this type of testing, but nonetheless if they frame their paper as testing for the existence of tuning curves, then certainly there should be an attempt to look at responses to stimuli varying along some dimension. A recent study by Suthana et al (2015), for example, identified images to which neurons responded, and then in a subsequent recording session, presented images of varying similarity to the images that initially elicited a response. The focus of that paper was not to assess the hippocampal tuning curve idea, but nonetheless the methodology used is more

similar to what I expected to see in the current study upon reading the framing and aims. The only aspect of the current study that seems to be at least somewhat well-suited to addressing the graded response idea is the use of an internet-based association rating between pairs of stimuli. By looking at neuronal responses to pairs of stimuli that range in association strength, the authors demonstrated that there is indeed a graded pattern, such that responses are very similar when the association between two stimuli is strong, and less so when the association is weaker. This finding, however, seems to be at odds with how they frame their results - the topic of my second point.

We agree with the framing proposed by the reviewer (a similar comment was also made by reviewer 3) and have changed the manuscript accordingly. As the reviewer suggested, we now frame the paper asking how hippocampal neurons represent associated stimuli, which is much more specific and to the point, toning down claims of no graded coding, as typically found in cortex. In particular, we now present the results in terms of the mechanism of 'neural unitization' as the mean for encoding long-term associations in the human medial temporal lobe. The Introduction and Discussion have been rewritten accordingly. To further support this idea, we have also included new analyses in the Results section (further details can be found in the reply to the next comment).

2. The authors interpret their findings as refuting the idea that hippocampal neurons exhibit graded responses similar to a tuning curve, and instead conclude that, when neurons respond to more than one stimulus, they respond equally strongly in a "unitized" manner. However, they also highlight that the more closely related two stimuli are, the more similar the neuronal responses to these images - which seems to imply a graded response. I realize that the latter results pertain to responses across many different units rather than within the same neuron, but nonetheless, it is challenging to reconcile these two ideas. These mixed messages are readily apparent in the abstract: "In contrast to the graded responses ubiquitously observed in cortex, we found that most of these neurons exhibited no differences in their spike and local field potential responses to the individual stimuli. Moreover, the similarity of the neural responses correlated with the degree of association between stimuli." How do the authors relate the latter finding to their conclusion that neuronal responses are not graded along some feature dimension (e.g., semantic relatedness)?

As suggested by the reviewer, we have now edited the manuscript to avoid contradictions in our message. Particularly, we have now clarified the idea that associated stimuli show unitized responses in a non-graded way, and that the responses that are not unitized (which is a small set) are mainly related to non-associated stimuli, and are indeed responsible for explaining the overall significant correlations that we observed in Fig. 6. Specifically, we have rewritten the section that is now called "Unitization as a mechanism for encoding association in the human MTL" and included further analyses to support our claims.

Particularly, we first showed that the results obtained in Fig. 6A, where we found a significant difference between AS_{R-R} and AS_{R-NR} , could not be attributed to semantic categorization, stimulus familiarity or visual similarity (see the response to comment number 2 from Reviewer 1), and are therefore better explained with the degree of association between responsive stimuli.

Second, as suggested by reviewer 3, we correlated the association strength for pairs of responses and non-responses (i.e. AS_{R-NR}) with the difference in strength and LFP latency. Strength, defined as spike count in a certain period of time, can be defined for responsive and non-responsive stimuli. LFP responses are seen for most stimuli (whether or not they are responsive in terms of spikes) so a latency can also be defined. However, this is not the case for the spike latency, as most non-responsive stimuli will not elicit a neural response from which a latency can be properly defined (e.g. in most cases the instantaneous firing rate will not even cross the threshold that is required to define the spike latency). The results can be seen in the following figure, which is also Supplementary Fig. 9.

There was no correlation between the association strength and the (normalized) strength difference ($r = 0.01$, $p = 0.74$), with significant differences when compared with the correlation for R-R pairs (Fisher Z transformation, $p = 9.2 \times 10^{-3}$). The same was the case for the LFP latency, with no correlation for R-NR ($r = -0.05$, $p = 0.08$), while being significantly different to the one for R-R (Fisher Z transformation, $p = 1.5 \times 10^{-2}$).

Finally, we observed that by discarding the largest quartile in terms of strength, spike and LFP latency difference, the resulting correlations in the remaining set vanished (strength: $n=104$, Pearson correlation, $r = 0.03$, $p = 0.77$; spike latency: $n=104$, Pearson correlation, $r = -0.05$, $p = 0.63$) or were reduced (LFP latency: $n=93$, Pearson correlation, $r = -0.15$, $p = 0.15$). Moreover, a comparison of the correlation for these subsets and the ones associated for R-NR pairs showed no significant differences for either strength (Fisher Z transformation, $p = 0.85$) or LFP latency (Fisher Z transformation, $p = 0.23$).

These results show that, while the associated stimuli are unitized, the non-associated and non-unitized stimuli are indeed responsible for explaining the overall significant correlations that we observed.

3. Finally, I found the analyses pertaining to Figures 1 and 2 to be a bit circular. Specifically, "multi-responsive" neurons were defined as those that met criteria for responding strongly to more than one stimulus. If a given neuron is defined as "multi-responsive" because it responds strongly to both A and B, it seems circular to then perform statistical analyses showing that the response to these two stimuli was very similar. It seems important, when creating the criteria for determining whether neurons are responsive, to use liberal thresholds that would then allow one to later test for differences in responsiveness. If, instead, a conservative set of criteria is used, then by definition the response to both stimuli has to be very strong; not surprisingly, when one then compares these responses, they are similar. Figure 2b helps to address this issue somewhat by showing the distribution of response strength for responsive units, but this analysis seems circular as well, in that it is comparing distributions for stimuli that did and did not elicit responses. Again, not surprisingly, when these distributions are compared, the distribution for "responses" has mostly high strength responses, and the distribution for "non-responses" has mostly low strength responses. It would be more helpful to see the see this distribution without the separation between responses and non-responses.

The figure below shows the response strength for all multi-responsive neurons (each color denotes a neuron). As we see in the figure, it is not necessarily the case that responses crossing a relatively conservative threshold will be similar - i.e. two responses could cross a

given threshold but still be very different from each other, as shown with the example of Fig S8b and particularly when comparing the normalized responses of the different neurons (compare for example, responses of Fig 1a with those of Fig S7). There is in fact a wide range of response strengths crossing the responsiveness criterion, but note that responses of the same neuron tend to cluster together because they are unitized. Comparing response differences within and between neurons is in fact what was quantified in Figure 2A (2B in the new version of the manuscript). In particular, if the finding of unitization would be due to the use of a conservative threshold for defining responsiveness, then one would expect similar differences between responses of the same neuron compared to the ones of other neurons, which was not the case (note that the responsiveness criteria is normalized by the baseline firing of each neuron, so comparing responses across neurons is fair). To clarify this point, we have included the figure below as Fig 2A in the new version of the manuscript.

Following the reviewer's suggestion, we have also repeated the main analyses considering more liberal (or conservative) thresholds. In particular, we show that similar results are obtained when considering 4, 5 and 6 standard deviations above the baseline activity (5 is the one we chose in the manuscript). Choosing a lower threshold of 2.5 s.d. gave different results since in this case we introduced many false positives that did not show responses. For details, see the response to comment number 3 from Reviewer 1.

Finally, we show below the distribution of response strength of Figure 2B (2C in the revised manuscript) without splitting between responsive and non-responsive stimuli, as requested by the reviewer.

The problem of not splitting the distributions is that the number of non-responsive stimuli is much larger than the number of responsive stimuli and therefore the strength distribution for responsive stimuli (at the right of the plot) is pushed down. Then, it is harder to determine if the stimuli with intermediate strengths are (non-unitized) responses or non-responses that show (by chance) a larger strength (as some of the exemplars presented in response to comment number 3 from Reviewer 1). For this reason, we prefer to leave the figure splitting between responsive and non-responsive stimuli in the manuscript. Note that the key point is not that we find that responsive stimuli have larger strength – this is indeed circular – but rather that the distribution is close to binary, with relatively few intermediate responses or something like a Gaussian tuning peaking at the maximum response. We have clarified this in the main text.

Minor concerns:

1. The term “unitization” has clear meaning in the memory literature – the process of creating an integrated representation that includes all elements/items of an association. If space allows in the Introduction, it would be useful to include a definition and a few references to prior studies of unitization to familiarize readers with this concept.

In the introduction we explain what we mean by ‘neural unitization’ the first time we mention this concept – i.e. “neurons that responded equally to the different stimuli eliciting significant responses, or in other words, if a neuron fired to more than one stimulus, the responses to

these stimuli were indistinguishable from each other". We agree with the reviewer that this can be mixed with the notion of 'unitization' used in Psychology and we have therefore included the following paragraph in the Discussion to avoid any confusion with the terminology:

"The concept of 'unitization' has been largely used in the Psychology literature (Graf and Schacter, 1989; Mayes et al., 2007; Diana et al., 2008; Murray and Kensinger, 2013; Parks and Yonelinas, 2015). In this sense, unitization involves representing previously separate items as a single entity. In our work, we use the concept of 'neural unitization' to refer to something different, namely whether different stimuli can be discriminated based on their neural responses, but without implying whether such stimuli can be discriminated or not at the behavioral level."

2. I found the term "responsive pairs" of stimuli to be confusing. The stimuli themselves weren't responsive, but rather, these stimuli elicited responses from neurons. When reading the manuscript, it was at times challenging to understand what "responsive pairs" referred to (i.e., pairs of stimuli, pairs of responses, etc.). To reduce this confusion, I suggest instead using the term "response-eliciting pairs" of stimuli.

We have followed the reviewer's suggestion and edited the manuscript accordingly.

Reviewer #3 (Remarks to the Author):

This manuscript reports single unit data from a modest number of neurons in human amygdala and hippocampus. The main findings are that some units were responsive to multiple stimuli, and that key features of the responses to the driving stimuli were similar and showed modest (but significant) correlations with an association strength metric. From these data, the authors speculate that the MTL neural code for these pairs/sets of stimuli is 'unitized'.

While the questions addressed are of potential interest, there are a number of dimensions of the manuscript that raise concerns. The main concerns are detailed below, with the hope that they will be helpful to the authors.

We thank the reviewer for the feedback and address the concerns below.

1) The manuscript's framing is ineffective. First, the Introduction frames the question around the broad issue of what is the nature of the neural code in the MTL. This is such a vast question, and unfortunately the Introduction does not situate the work within the rich literature on this topic (e.g., complementary learning systems theories; the wealth of data on how stimulus and environmental stimuli that vary is similar given rise to continuous vs. sigmoidal responses that bear on claims about pattern separation vs. pattern completion; the animal and human literature on memory integration, unitization, and integrative encoding; etc). Moreover, the notion of an 'association' is introduced in the Abstract and Introduction, but it is not until well into the results that the reader comes to realize that the work relates to the sub-literature on the relationship between pre-existing knowledge/concepts and MTL coding.

I would recommend that the authors completely re-frame the Abstract and Introduction, narrowing it rather to the set of questions that has emerged about the role of pre-existing associative knowledge and MTL coding. I would then conclude the Introduction with a set of hypotheses to be tested that are specific to this aspect of the experiment.

We have followed the reviewer's suggestion and edited the manuscript accordingly (see the response to comment number 1 from Reviewer 2).

2) Stimuli during the follow up session were presented in a 'pseudorandom order'. What is meant by 'pseudorandom' (vs. random) is unclear. More critically, given evidence of rapid sequential/statistical learning effects within the hippocampus, it seems important to compute the first and second order transition probabilities (and perhaps even longer trial history effects) between

stimuli and to then examine whether the probability of a 'unitization' effect also relates in any way to the transition probabilities. Note that by including transition probabilities as covariates, this may also increase sensitivity to pre-experimental associative effects.

We have now clarified the experimental design in the Methods section, which now reads:

*"a set of about 100 stimuli were presented, six times each in pseudorandom order using a block design (i.e. if N stimuli are used in the session, all stimuli will be shown once, in random order, after the first block of N trials, twice after the first 2*N trials, etc.)"*

Particularly, the order of the N stimuli within each block was randomized using the command *randperm(N)* in Matlab. Therefore, by construction, the transition between stimuli will have a flat distribution, preventing them from having an effect in the way that stimuli are associated. Moreover, as we mentioned in the Discussion, the associations we studied in this work are pre-existent. If they were actually arbitrarily generated by the way in which stimuli were shown during the experimental paradigm, our web-based association score would not be able to reflect those associations, and the results from Figure 6 would change completely.

3) As the authors state, the association score that was computed is a measure of pre-experimental semantic relatedness, rather than a measure of episodic association.

- Note that this effect appears rather weak. Why use a one-sided paired sign test? The fact that this is a one-sided test should be stated in the results text; currently this only appears in the figure caption.

The weakness of the effect is partly due to the low number of multi-responsive units ($n = 37$) and stimuli used in the follow-up sessions. However, our result is in line with the one obtained in De Falco et al (2016), where a much larger dataset was used to show that multi-responsive units tend to fire in response to associated stimuli. Still, we performed controls on semantic categories, stimulus familiarity and visual similarity (see the response to comment number 2 of Reviewer 1).

The one-sided test is used to test that R-R vs R-NR are not just different, but that the former is larger than the latter and therefore a one-sided test is appropriate. As requested by the reviewer, we have now included this information (i.e. whether tests were one or two sided) throughout the main text.

- Given that this score measures pre-experimental relatedness, what accounts for why the multi-responsive neurons did not originally respond to both members of a pair of stimuli that ultimately were classified as R-R during the follow up period of the experiment. Were all stimuli not included during stimulus selection? If not, how were

additional stimuli selected for inclusion in the follow up phase of the experiment? If so and if these pairs were already linked in semantic memory, why was repeated repetition required to elicit a similar response in the small population of amygdala and hippocampal neurons identified as multi-responsive here?

We have not said that the multi-responsive units did not elicit their responses during the initial screening session. However, we argue that such session would not be adequate to properly compare the neural responses to different stimuli due to the limited number of trials per stimulus (6). In fact, Supplementary Figures S2 and S4 study the effect of the number of trials used to estimate strength and latency, showing that 6 trials are not enough to obtain stable estimates of either strength or spike latency.

Regarding the stimulus selection for the follow-up session, we explained in the Methods that we included all the responsive stimuli from the initial screening session, and a few more stimuli to complete the set when not enough responses were found. Still, we showed above (see response to comment number 3 from Reviewer 1) that the stimuli that were left out for the follow-up session did not exhibit a different firing than the non-responsive ones already included in the follow-up set.

- It would be helpful to examine the temporal profile over which the multi-responsive units came to demonstrate such responses. That is, was the effect evident immediately at the start of follow up or did it take time to emerge. If immediately at the outset of follow up, did this differ in any way from the stimulus selection phase of the experiment?

As we mentioned in the reply to the previous comment, the responses in the initial screening session do not have enough trials to properly compare them. Still, considering our experimental design with an initial screening session and a follow-up session, all the responsive stimuli from the screening session were included in the follow-up session. Moreover, the exemplary raster plots presented in this work (Figures 1a, 3a, 5, S7, and S8) show that the responses could be observed since the beginning of the session (first trial on top).

As requested by the reviewer, we computed the spike count per trial for each response and subtract the baseline count to get a normalized strength of response as a function of trial number. The following figure shows the grand average across all the 165 responses (blue) and the exponential fit (red).

Clearly, the response strength is being reduced after a few trials (due to repetition suppression effects), but it reaches a steady-state after ~6 trials, with the response staying significantly higher than baseline (which would be zero in the y axis). Since we used an average of 29 trials in our dataset, our strength estimates would be reliable, and any repetition suppression effect at the beginning of the session would be negligible.

- For the correlations plotted in Figs 6b-d, please also plot the corresponding correlations for the R-NR pairs, and please test and report whether the strength of the R-R correlations significantly differed from that of the R-NR correlations? Moreover, related to the point about the effect size, given the small amount of variance explained by the association score, the following conclusion seems to overstate the data: "the degree of association appears to be the key metric underlying the neuronal coding in the MTL". No other behavioral or stimulus dimensions were explored during data analysis, and so such a conclusion does not seem justified. Note also, given that the hippocampus can rapidly form relational/associative/conjunctive representations of previously unassociated pairs stimuli (as is stated in the Discussion), such a statement (which appears to apply to pre-existing associations) appears inconsistent with a vast literature.

As suggested, we have now included the correlations with AS_{R-NR} (see the response to comment number 2 from Reviewer 2). We have also removed the claim "the degree of association appears to be the key metric underlying the neuronal coding in the MTL". Instead, as explained above, we have now clarified the idea that associated stimuli show unitized responses in a non-graded way, and that the responses that are not unitized (which is a small set) are mainly related to non-associated stimuli and are indeed responsible for explaining the overall significant correlations that we observed.

Also, we agree with the reviewer that the hippocampus has a unique machinery to rapidly associate arbitrary stimuli (and we have acknowledged this in the Discussion). In fact, the work from Ison et al (2015) showed this in the human MTL. However, what we have shown in this work is that there is a different way of encoding new vs. long-term associations. While new associations showed graded responses (i.e. the neuron's graded firing was enough to discriminate between the item originally coded by the neuron and the one that was associated with it), the long-term associations we studied in this work showed unitized responses.

4) The notion of 'unitization' has pre-existing meaning in the literature on MTL coding (e.g., see work from Ranganath and colleagues). In that past work, the 'unitization' hypothesis led to articulation of clear behavioral criteria had to be met for two stimuli to be viewed as 'unitized'. In the present work, not such criteria are specified. As such, it is unclear what the authors mean by 'unitization' here, nor whether there is independent behavioral evidence supporting such a claim (beyond the similar neural responses reported herein).

Following the reviewer suggestion, we have added a paragraph in the Discussion to clarify that the proposed mechanism of 'neural unitization' should not be confused with the idea of 'unitization' commonly used in the Psychology literature. Please refer to the response to the minor concern number 1 from Reviewer 2 for further details on how we have clarified this issue in the revised manuscript.

Minor comments

a) From the methods, it is unclear what percentage of the units in the hippocampus and in the amygdala failed to meet the criteria for being 'responsive to a certain picture'.

We have now clarified in the Methods section that the 81 responsive units came from a total of 609 recorded units during the follow-up sessions.

b) Please report the number of units in hippocampus and amygdala that were responsive in each subject. Moreover, while the number of multi-responsive units in each region was small, it seems important to at least provide a qualitative statement as to whether there were regional differences of any kind in the degree of response similarity as a function of association strength. If no differences are evident, do the authors argue that the amygdala codes for concepts in the same way that they are arguing for concept coding in hippocampus?

As individual subjects performed multiple sessions, we report here the number of units on each session

Session number	1	2	3	4	5	6	7	8	9	10	11	12	13	14	15	16	17	18	19	20	21	Total
# neurons in Hippocampus	12	9	3	1	1	2	1	1	1	1	1	5	2	1	1	6	5	5	5	4	3	70
# neurons in Amygdala	0	0	0	0	0	0	0	0	0	0	3	0	0	0	0	3	2	2	1	0	0	11

From the 37 multi-responsive units, only 7 were from the amygdala, preventing us from splitting our analyses according to the different regions we recorded from due to low statistical power. However, our group has published several papers analyzing regional differences (or lack of them) within the human MTL. Particularly, it has been shown that there are no differences between hippocampus and amygdala in terms of percentage of responsive units (Kornblith et al, Current Biology, 2017), spike latency (Mormann et al, J Neurosci, 2008), and more importantly to our results, the tendency of neurons to fire to associated concepts (De Falco et al, Nat Comm, 2017). Therefore, it is unlikely that the mechanism we described in this paper would be valid only for the hippocampus.

c) The methods and results reporting are rather confusing; this relates, in part, to the challenging framing of the manuscript (as I read the results and methods, I found myself repeatedly trying to figure out why particular analyses were being performed and over which aspects of the data). As one critical example, when stating that 37 of the 81 'responsive units' were 'multi-responsive', was this during the initial recording/stimulus selection period or during the follow up repeated presentation period? I assume the latter, but this is difficult to discern early on when reading the methods and results.

Regarding the particular example mentioned by the reviewer, we clarified in the Introduction and the first section of the Methods ("Subjects and recordings") that the data reported in the manuscript comes from the follow-up sessions. Overall, we have tried to be as clear as possible in explaining all the methodology we have used in our work. Particularly, we have now changed the framing of the paper to make it clearer and focused on the idea of neural unitization as a mechanism for encoding long-term associations.

Reviewers' comments:

Reviewer #1 (Remarks to the Author):

The authors did a remarkable job at addressing many of my initial concerns and have significantly improved on the paper. In particular, they addressed my concerns about interpretation of the findings in relation to memory vs. categorization or familiarity-related processes. They also answered concerns on the possibility of pre-selection bias.

Overall, I believe that this is an important study and I would like to see it published. I also went over the comments and responses to the other reviewers and believe that the authors have done a thorough job at addressing their concerns. My only final recommendation would be to remove the statement "In contrast to the graded responses ubiquitously observed in cortex, " from the abstract since this study did not involve direct evaluation or comparison to other cortical neurons. I believe that making a historical comparison here would not be appropriate.

Reviewer #2 (Remarks to the Author):

The authors were very responsive to reviewer concerns, and I believe that the manuscript is much stronger as a result. The framing is now much more focused, and thus a better fit for the experiment that was performed. Additionally, several new analyses help to attenuate concerns regarding selection bias/double-dipping. Finally, added clarification of the authors' use of "unitization" is helpful as well. Although the manuscript is much improved, I believe that addressing the following minor points will be beneficial, as well:

(1) On the first page, the authors state: "In this work, we sought to compare the neural responses of well-learned associated stimuli to test if they remain graded as when the associations were originally learned." This phrasing is misleading, making it appear as though the study looks at two time points (learning of associations and consolidated associations), whereas only consolidated associations were examined.

(2) In the same paragraph, the authors state "... we first identified single cell responses in the human MTL, and then presented the stimuli eliciting responses many times...". Given my familiarity with the study design, I understand what the authors are saying, but I think the following may be more useful to readers: "... we first identified stimuli to which single cells in the human MTL responded, and then, in follow-up sessions, presented these response-eliciting stimuli many times..."

(3) Although the authors changed several instances of "responsive stimuli" to "response-eliciting stimuli", many instances of the former still remain (see, e.g., lines 137-144). As stated in my prior review, the stimuli themselves are not responsive; rather, they elicit responses from neurons.

Reviewer #3 (Remarks to the Author):

This revised manuscript effectively addressed a number of the points raised during initial review. While strengthened, the manuscript's framing in the Abstract and Introduction continue to strike this reviewer as needing further clarification and sharpening. The critical nature of the associations being assayed here and the authors' notion of unitization are not well framed at the outset. I continue to think that leading (in the Abstract and Introduction) with the question of how well-learned ("semanticized") associations are represented in the MTL would better position the work at the conceptual/theoretical level. In addition, a few methods/results details also are missing (as noted below).

Specific comments

- Abstract: It will be unclear to the reader, at this point in the manuscript, what is meant by “for largely associated stimuli”. It could mean --- “for strongly associated stimuli” or “primarily for associated stimuli”.
- Abstract: The construct of “neural unitization” and “the unitized code” is introduced in the Abstract, but it will be unclear to the reader what is meant by this construct at a representational level. At present, it is only defined by three features of the response — neurons that respond to more than one stimulus, that their activity properties do not differ across these stimuli, and that the stimuli driving such responses tend to be associated. These properties, however, do not specify for the reader what is meant by “unitization” when it comes to the framing questions about the nature of the neural code.
- The Introduction’s first paragraph should make clear that the present focus is on pre-existing associations rather than on the encoding of new (one-shot) arbitrary associations. This important conceptual point was raised during initial review; the revisions are helpful, but do not fully resolve this ambiguity for the reader.

Confusion about the questions of interest and the nature of the experiment will be further driven by text later in the Introduction: “we sought to compare the neural responses of well-learned associated stimuli to test if they remained graded as when the associations are originally learned.” This implies that there will be a learning phase that compares neural responses to new arbitrarily associated stimuli to the responses to these same stimuli after they’ve been studied together 25-30 times. But this is not what was done. Rather, all that was done is to examine the response to individual stimuli appearing 25-30 times and seeing if neurons that responded to one stimulus also responded to others, if those responses statistically differed, and if those responses relate to pre-experimental associative strength. Respectfully, the framing of the manuscript continues to intermix and confuse multiple important distinct concepts.

- My apologies if I missed this spelled out somewhere, but I was unable to determine the mean number of picture pairs/sets and the range of pairs/sets across patients that elicited activity in the same neuron(s) during the screening phase. As I understand results reporting, there were 208 response-eliciting pairs of stimuli identified across the 6 patients. This implies that there was a mean of 34.7 response-eliciting pairs of stimuli per patient, yet there were only about 15 stimuli that elicited any response in a given patient (as I understand it). How many of the pictures then fell into the response-eliciting category and how many fell into the other category (mean and range would be helpful here)? In addition, while computable from such numbers, perhaps the authors can state what percentage of the stimuli that elicited any response fell into response-eliciting pairs and what percentage of stimuli did not fall into such a pair?

This design issue is fundamental, because depending on the number of pictures that fell into the response-eliciting pairs category, this has implications for data interpretation — if there are few stimuli that elicit these responses, then one might argue that the reported ‘unitization’ pattern is atypical rather than the norm. By contrast, if most of the stimuli that elicited a response from any of the neurons fell into the eliciting pairs category, then this might imply that this is the norm for such overlearned (semanticized) associations.

- While I appreciate the authors adding more detail about their pseudorandomization approach, their comments regarding the potential contributions of rapid statistical learning effects appeared to focus largely on the screening phase. However, during the critical data collection phase (where stimuli appeared 25-30 times) there is reason to believe that first order and second order learning effects could be present (given that only about 15 pictures appeared in each of the blocks). It seems likely that with the pseudorandomization approach implemented, for a given patient there

will be some picture pairs with higher transition probability than others. By exploring the transition distributions and, assuming the pairs differ in their first and second order transitions, by examining whether there are rapid learning effects, this would allow the authors to (a) remove such effects from the present analyses, as such effects would be 'noise' that reduces the measured effect sizes, and (b) compare whether newly acquired arbitrary associations might reveal MTL responses that are the primitives [earliest substrates] for the putative unitized representations explored here. These primitives could come about through big-loop recurrence / integrative encoding and could constitute the building of so-called successor representations that, from the authors' current framework, may be ultimately unitized. The latter would broaden the impact of the work, linking/contrasting new associative learning effects with long-term multi-shot/semantic effects. At present, the potential similarities/differences between new associative memory effects and well-learned long-term effects can only be made indirectly, by comparing the present work to prior studies.

- Thank you for providing the number of responsive units per patient per MTL region (hippocampus vs. amygdala). Unless I missed it, please add this informative table to the Supplement and please also note that of the 37 multi-responsive units, 30 were from hippocampus. This will make clear for the reader that the vast majority of the critical data in the present study come from hippocampus.

RESPONSE TO THE REVIEWERS' COMMENTS

We thank the reviewers for the very useful feedback, which we believe have helped us to improve our manuscript.

Reviewer #1 (Remarks to the Author):

The authors did a remarkable job at addressing many of my initial concerns and have significantly improved on the paper. In particular, they addressed my concerns about interpretation of the findings in relation to memory vs. categorization or familiarity-related processes. They also answered concerns on the possibility of pre-selection bias.

Overall, I believe that this is an important study and I would like to see it published. I also went over the comments and responses to the other reviewers and believe that the authors have done a thorough job at addressed their concerns. My only final recommendation would be to remove the statement "In contrast to the graded responses ubiquitously observed in cortex" from the abstract since this study did not involve direct evaluation or comparison to other cortical neurons. I believe that making a historical comparison here would not be appropriate.

We appreciate the comments from the reviewer. We have removed the statement from the Abstract as suggested.

Reviewer #2 (Remarks to the Author):

The authors were very responsive to reviewer concerns, and I believe that the manuscript is much stronger as a result. The framing is now much more focused, and thus a better fit for the experiment that was performed. Additionally, several new analyses help to attenuate concerns regarding selection bias/double-dipping. Finally, added clarification of the authors' use of "unitization" is helpful as well.

We appreciate the comments from the reviewer.

Although the manuscript is much improved, I believe that addressing the following minor points will be beneficial, as well:

(1) On the first page, the authors state: "In this work, we sought to compare the neural responses of well-learned associated stimuli to test if they remain graded as when the associations were originally learned." This phrasing is misleading, making it appear as though the study looks at two time points (learning of associations and

consolidated associations), whereas only consolidated associations were examined.

Following the reviewer's suggestion, we have edited this sentence, which now reads: "In this work, we sought to compare the neural responses of well-learned associated stimuli to gain insights on the neural code underlying the long-term representation of associations in the human MTL."

(2) In the same paragraph, the authors state "... we first identified single cell responses in the human MTL, and then presented the stimuli eliciting responses many times...". Given my familiarity with the study design, I understand what the authors are saying, but I think the following may be more useful to readers: "... we first identified stimuli to which single cells in the human MTL responded, and then, in follow-up sessions, presented these response-eliciting stimuli many times..."

We have edited this sentence as suggested by the reviewer.

(3) Although the authors changed several instances of "responsive stimuli" to "response-eliciting stimuli", many instances of the former still remain (see, e.g., lines 137-144). As stated in my prior review, the stimuli themselves are not responsive; rather, they elicit responses from neurons.

We apologize for not having edited all the instances where we used the term "responsive stimuli". We have conducted now a more thorough search across the manuscript and replaced the term as suggested.

Reviewer #3 (Remarks to the Author):

This revised manuscript effectively addressed a number of the points raised during initial review. While strengthened, the manuscript's framing in the Abstract and Introduction continue to strike this reviewer as needing further clarification and sharpening. The critical nature of the associations being assayed here and the authors' notion of unitization are not well framed at the outset. I continue to think that leading (in the Abstract and Introduction) with the question of how well-learned ("semanticized") associations are represented in the MTL would better position the work at the conceptual/theoretical level. In addition, a few methods/results details also are missing (as noted below).

Specific comments

- Abstract: It will be unclear to the reader, at this point in the manuscript, what is meant by "for largely associated stimuli". It

could mean --- "for strongly associated stimuli" or "primarily for associated stimuli".

We have now removed this statement from the Abstract.

- **Abstract:** The construct of "neural unitization" and "the unitized code" is introduced in the Abstract, but it will be unclear to the reader what is meant by this construct at a representational level. At present, it is only defined by three features of the response -- neurons that respond to more than one stimulus, that their activity properties do not differ across these stimuli, and that the stimuli driving such responses tend to be associated. These properties, however, do not specify for the reader what is meant by "unitization" when it comes to the framing questions about the nature of the neural code.

We have edited the Abstract and avoided the use of the term "unitization", which is defined in the introduction.

- The Introduction's first paragraph should make clear that the present focus is on pre-existing associations rather than on the encoding of new (one-shot) arbitrary associations. This important conceptual point was raised during initial review; the revisions are helpful, but do not fully resolve this ambiguity for the reader.

The first paragraph is a general introductory statement about the MTL role in memory and the encoding of associations, but to avoid any confusion, we have edited this paragraph and now refer to the 'encoding of associations' rather than to the 'rapid and effortless formation of associations', which may give the incorrect impression that our paper deals with memory formation.

Confusion about the questions of interest and the nature of the experiment will be further driven by text later in the Introduction: "we sought to compare the neural responses of well-learned associated stimuli to test if they remained graded as when the associations are originally learned." This implies that there will be a learning phase that compares neural responses to new arbitrarily associated stimuli to the responses to these same stimuli after they've been studied together 25-30 times. But this is not what was done. Rather, all that was done is to examine the response to individual stimuli appearing 25-30 times and seeing if neurons that responded to one stimulus also responded to others, if those responses statistically differed, and if those responses relate to pre-experimental associative strength. Respectfully, the framing of the manuscript continues to intermix and confuse multiple important distinct concepts.

Following the reviewer comment (and a similar comment of reviewer 2), we have edited this paragraph to avoid giving the impression that our study included a learning phase. In particular, the first sentence of the paragraph now reads: "In this work, we sought to compare the neural responses of well-learned associated stimuli to gain insights on the neural code underlying the long-term representation of associations in the human MTL."

- My apologies if I missed this spelled out somewhere, but I was unable to determine the mean number of picture pairs/sets and the range of pairs/sets across patients that elicited activity in the same neuron(s) during the screening phase. As I understand results reporting, there were 208 response-eliciting pairs of stimuli identified across the 6 patients. This implies that there was a mean of 34.7 response-eliciting pairs of stimuli per patient, yet there were only about 15 stimuli that elicited any response in a given patient (as I understand it). How many of the pictures then fell into the response-eliciting category and how many fell into the other category (mean and range would be helpful here)? In addition, while computable from such numbers, perhaps the authors can state what percentage of the stimuli that elicited any response fell into response-eliciting pairs and what percentage of stimuli did not fall into such a pair?

We thank the reviewer for this comment, as we now realize the explanation of the experiment and its rationale was not clear enough in the introduction. As mentioned above, we have edited the first two paragraphs to avoid giving the impression our study includes a learning phase, and we have also edited the final paragraph to clarify the rationale of our study. In particular, we now explain in more detail the rationale for having a first screening session followed by another experimental session where pictures were shown many more times (and that all the analyses reported corresponds to these follow-up sessions).

Following the reviewer's comment, in the Results section (2nd paragraph) we have added the information of how many responses we had on the multi-responsive neurons (19 units responded to 2 pictures, 5 units to 3 pictures, 6 units to 4 pictures, and 7 units to 5 or more pictures), and how many response-eliciting pairs were found per neuron (mean \pm s.d = 5.6 \pm 9.9).

This design issue is fundamental, because depending on the number of pictures that fell into the response-eliciting pairs category, this has implications for data interpretation -- if there are few stimuli that elicit these responses, then one might argue that the reported 'unitization' pattern is atypical rather than the norm. By contrast, if most of the stimuli that elicited a response from any of the neurons fell into the eliciting pairs category, then this might imply that this is the norm for such overlearned (semanticized) associations.

There are two questions we should separate. First, how typical it is to get neurons firing to more than one stimulus, which is about 45%, as from 81 responsive units, 37 responded to two or more things (as stated in the Results section). Second, for those neurons responding to two (or more) stimuli, how typical it is that the responses to these two (or more) stimuli are the same. This is what we report in our paper.

To be more concrete. From the 37 'multiresponsive' units, we have 208 pairs of responses to be compared (1 pair for a neuron with 2 responses, 3 pairs for a neuron with 3 responses, etc). All our results are reported as a fraction of these 208 pairs (or the 37 multiresponsive neurons). In other words, what we argue is that if a neuron responds to two or more stimuli, in about 80% of the cases the response to these stimuli will be statistically the same (irrespective of how likely it is to find responses to more than one stimulus, which depends on the number of stimuli presented in the initial screening session, as the neuron may have responded to other stimuli that were not used).

- While I appreciate the authors adding more detail about their pseudorandomization approach, their comments regarding the potential contributions of rapid statistical learning effects appeared to focus largely on the screening phase. However, during the critical data collection phase (where stimuli appeared 25-30 times) there is reason to believe that first order and second order learning effects could be present (given that only about 15 pictures appeared in each of the blocks). It seems likely that with the pseudorandomization approach implemented, for a given patient there will be some picture pairs with higher transition probability than others. By exploring the transition distributions and, assuming the pairs differ in their first and second order transitions, by examining whether there are rapid learning effects, this would allow the authors to (a) remove such effects from the present analyses, as such effects would be 'noise' that reduces the measured effect sizes, and (b) compare whether newly acquired arbitrary associations might reveal MTL responses that are the primitives [earliest substrates] for the putative unitized representations explored here. These primitives could come about through big-loop recurrence / integrative encoding and could constitute the building of so-called successor representations that, from the authors' current framework, may be ultimately unitized. The latter would broaden the impact of the work, linking/contrasting new associative learning effects with long-term multi-shot/semantic effects. At present, the potential similarities/differences between new associative memory effects and well-learned long-term effects can only be made indirectly, by comparing the present work to prior studies.

The reviewer argues that some associations may have been given by the temporal proximity in the presentation of certain stimuli (e.g. if stimulus 5 was typically presented close to

stimulus 1, a neuron responding to 1 may start responding to 5 as well). As mentioned in the previous revision, we expect this not to be the case, given that the stimuli were presented in pseudorandom order on each block. Still, to rule this out, for each pair of stimuli presented on each experimental session we computed the number of interleaving stimuli between them and obtained the mean of that distribution (as expected, this value was always close to half the number of stimuli presented in the session). Then, for each session, we considered the pairs where both stimuli elicited responses (R-R) and those where only one did it (R-NR). If the responsive pairs we analysed were the product of rapid learning effects appearing during the task, the distribution of interleaving stimuli between R-R pairs should be smaller than for R-NR pairs. Yet, we found no evidence supporting this in any of the sessions (ranksum test, $p > 0.05$ in all cases).

Moreover, we used the mean of the interleaving stimuli distribution computed for all the 208 R-R pairs to define the first and fourth quartile, i.e. the quarter of the pairs with the smallest and largest number of interleaving stimuli. When comparing these subsets of pairs, we found no difference in their association scores (ranksum test, $p = 0.97$), the normalized strength difference (ranksum test, $p = 0.13$), and the spike latency difference (ranksum test, $p = 0.44$).

Overall, we found no evidence to support that rapid learning effects due to proximity of stimulus presentation can explain the responses of the neurons.

• Thank you for providing the number of responsive units per patient per MTL region (hippocampus vs. amygdala). Unless I missed it, please add this informative table to the Supplement and please also note that of the 37 multi-responsive units, 30 were from hippocampus. This will make clear for the reader that the vast majority of the critical data in the present study come from hippocampus.

We have added the table in the Supplementary Material as requested.

REVIEWERS' COMMENTS:

Reviewer #3 (Remarks to the Author):

The authors have addressed all comments from the prior review phase.